# Lipoic acid rejuvenates aged intestinal stem cells by preventing age-associated endosome reduction

Gang Du[1,2], Yicheng Qiao[2], Zhangpeng Zhuo[2], Jiaqi Zhou[2], Xiaorong Li[2], Zhiming Liu[1], Yang Li[2] & Haiyang Chen[1,2,*]

## Abstract

The age-associated decline of adult stem cell function is closely related to the decline in tissue function and age-related diseases. However, the underlying mechanisms that ultimately lead to the observed functional decline of stem cells still remain largely unexplored. This study investigated *Drosophila* midguts and found a continuous downregulation of *lipoic acid synthase*, which encodes the key enzyme for the endogenous synthesis of alpha-lipoic acid (ALA), upon aging. Importantly, orally administration of ALA significantly reversed the age-associated hyperproliferation of intestinal stem cells (ISCs) and the observed decline of intestinal function, thus extending the lifespan of *Drosophila*. This study reports that ALA reverses age-associated ISC dysfunction by promoting the activation of the endocytosis–autophagy network, which decreases in aged ISCs. Moreover, this study suggests that ALA may be used as a safe and effective anti-aging compound for the treatment of ISC-dysfunction-related diseases and for the promotion of healthy aging in humans.

**Keywords** aging; alpha-lipoic acid; endocytosis; intestinal stem cell; longevity
**Subject Categories** Development; Membrane & Trafficking; Stem Cells & Regenerative Medicine

See also: **P Zhang and BA Edgar** (August 2020)

## Introduction

Organismal aging is characterized by a continuous decrease of the functional abilities of tissues and organs. In many vertebrate organs, resident adult stem cells, which possess high proliferative and differentiate capacities that compensate for cell loss, are responsible for both tissue homeostasis and organ functionality throughout the lifespan of the organism. This is particularly important in tissues with a high turnover rate, such as the intestinal epithelium (Guo *et al*, 2016). Although stem cells can be regarded as immortal, they are also subject to an age-associated decline of their self-renewal and differentiation properties (Schultz & Sinclair, 2016). Previous studies reported that damage accumulation in stem cells is closely linked to both organismal aging and age-related diseases such as cancer, inflammatory diseases, type 2 diabetes, and degenerative diseases (Rando, 2006; Apidianakis & Rahme, 2011; Li & Jasper, 2016; Holmberg *et al*, 2017). However, the detailed mechanisms that ultimately lead to this decline in stem cell function in response to aging still remain largely unknown. Identifying small molecular compounds that enhance the regenerative capacity of adult stem cells in model animals could not only yield drugs that promote healthy aging but could also provide additional insight into uncovering the mechanisms of this age-associated functional decline in stem cells.

Alpha-lipoic acid (ALA; 1,2-dithiolane-3-pentanoic acid, also known as thioctic acid) is an organosulfur-containing compound, which is present in all eukaryotic cells (Solmonson & DeBerardinis, 2018). In humans, ALA can be endogenously synthesized using intermediates from mitochondrial fatty acid synthesis type II, S-adenosylmethionine, and iron–sulfur clusters (Cronan, 2016; Shaygannia *et al*, 2018). In addition, as a vitamin-like substance, ALA can also be absorbed by digestion. As an essential cofactor for the energetic metabolism, multiple mitochondrial dehydrogenase complexes require ALA for catalysis, including pyruvate dehydrogenase, α-ketoglutarate dehydrogenase, and branched-chain ketoacid dehydrogenase (Solmonson & DeBerardinis, 2018). ALA has been reported to play critical roles in diverse biological processes, including the stabilization and regulation of mitochondrial multi-enzyme complexes, the elimination of reactive oxygen species (ROS), the oxidation of carbohydrates and amino acids, and the coordination of energetic metabolism (Park *et al*, 2014). Although healthy young humans can synthesize sufficient ALA to meet the body's needs, the level of ALA significantly declines with age, which is assumed to be linked to age-associated organic dysfunction (Hagen *et al*, 2002; Park *et al*, 2014). As a safe and natural ingredient, ALA has been widely administrated as a nutritional supplement to treat diverse age-associated diseases, including diabetes, obesity, diabetic polyneuropathy and retinopathy, atherosclerosis, hypertension, and Alzheimer's disease (Shay *et al*, 2009; Park *et al*, 2014; Salehi *et al*, 2019). In addition, it has been reported that ALA administration can reverse memory impairment and protect retinal pigment epithelial

---

1    Laboratory for Stem Cell and Anti-Aging Research, National Clinical Research Center for Geriatrics, West China Hospital, Sichuan University, Chengdu, China
2    Key Laboratory of Gene Engineering of the Ministry of Education, State Key Laboratory of Biocontrol, School of Life Sciences, Sun Yat-sen University, Guangzhou, China
    *Corresponding author. Tel: +86 028 85164205; E-mail: chenhy87@mail.sysu.edu.cn

cells in aged mice (Voloboueva *et al*, 2005; Farr *et al*, 2012). Although the mechanism of how ALA benefits aged tissues and organs remains largely unknown, these data indicate that ALA supplementation in elders may promote healthy aging.

Interestingly, recent studies have shown that ALA may also participate in the regulation of stem cell functions. ALA administration has been reported to reduce the loss of hematopoietic stem cells in G protein-coupled receptor kinase-depleted mice (Le *et al*, 2016). ALA can significantly increase the cardiac differentiation efficiency of embryonic stem cells (Shen *et al*, 2014). Moreover, ALA supplementation enhanced the therapeutic effect of mesenchymal stem cells in rats with cardiac injury (Abd El-Fattah *et al*, 2019). However, despite these clues, the role of ALA in the context of stem cell aging still remains unexplored.

Due to its simple organizational structure, straightforward genetic manipulation, and well-defined stem cell lineage, the *Drosophila* midgut has emerged as a suitable model system for the study of mechanisms underlying the age-related decline in stem cell function. Consequently, the *Drosophila* midgut can be used to identify potential strategies that enhance the regenerative capacity of adult stem cells. *Drosophila* intestinal stem cells (ISCs) specifically express Notch ligand Delta (Dl) and the transcription factor escargot (Esg), which reside in the basement membrane of the midgut epithelium. Here, ISCs proliferate to self-renew and produce progenitor cells (either enteroblasts [EBs] or enteroendocrine mother cells [EMCs], depending on the Notch activity). EBs further differentiate into absorptive enterocytes (ECs), and EMCs produce secretory enteroendocrine cells (EEs; Fig EV1A). The number of ISCs and progenitor cells is relatively small and remains stable in young and healthy midguts, while it increases several folds in response to aging (Biteau *et al*, 2008; Choi *et al*, 2008; Schultz & Sinclair, 2016). Furthermore, the differentiation capacity of ISCs and EBs showed a continuous decrease with aging (Cui *et al*, 2019). This leads to the accumulation of Esg and Dl expressing cells in the aged midgut (Biteau *et al*, 2008; Choi *et al*, 2008; Park *et al*, 2009). Multiple signaling pathways, such as c-Jun N-terminal kinase (JNK) signaling (Biteau *et al*, 2008), insulin signaling (Biteau *et al*, 2010; Cheng *et al*, 2014; Kao *et al*, 2015), mTOR signaling (Johnson *et al*, 2013), p38-MAPK signaling (Park *et al*, 2009), DGF/VEGF signaling (Choi *et al*, 2008), and ROS signaling (Hochmuth *et al*, 2011; Chen *et al*, 2017), have been reported to regulate ISC aging. Combined with notable transcriptional alteration, these signalings regulate the changes of biologic behaviors of ISCs during aging. This further disrupts the intestinal barrier and the acid–base balance of the digestive tract (Li *et al*, 2016). Interestingly, decreasing the number of *Esg*-positive (*Esg$^+$*) cells in the midgut (either by genetic manipulation or drug administration) significantly increased the lifespan of *Drosophila* (Gervais & Bardin, 2017). Therefore, the *Drosophila* midgut is an ideal model to investigate the function and the underlying mechanism of ALA in the regulation of the behaviors of stem cells upon aging.

Using *Drosophila* midgut as a model system enabled the disclosure of the role of ALA in the prevention of the functional decline of ISCs and the extension of the lifespan of *Drosophila*. This study reports that ALA increases *Drosophila* lifespan, regulates age-associated acid–base homeostasis, and prevents the age-associated hyperproliferation of ISCs through an endocytosis-mediated mechanism. Furthermore, this study suggests that ALA can be used as an effective and safe anti-aging compound to promote healthy aging in humans.

## Results

### Orally administered ALA rejuvenates aged ISCs

When *Drosophila* age, the ISCs in their midguts undergo a malignant increase of their proliferation rate and a decrease of differentiation efficiency (Biteau *et al*, 2008; Choi *et al*, 2008; Cui *et al*, 2019). This leads to the continuous accumulation of *Esg$^+$* cells (ISCs and their differentiating progenies) in the midguts of aged flies (Biteau *et al*, 2008; Choi *et al*, 2008; Cui *et al*, 2019). To disclose the regulation of the age-associated functional decline of ISCs by endogenous small molecules, the repressive effect on age-related *Esg$^+$* cell accumulation of 10 selected endogenously and *de novo* synthesized chemicals in *Drosophila* midguts was tested using an "*esg*-luciferase" reporter system (Figs 1A and B, and EV1B). This system is based on the luciferase reporter expression by *esg-GAL4*-driven *UAS-luciferase* in *Esg$^+$* cells, which allows the tracing and quantification of real-time changes of ISCs and their differentiating progenies in aging *Drosophila* midguts. Among these tested endogenous chemicals, ALA administration started at an intermediate age (26 days) and showed a most remarkable repressive effect of *Esg$^+$* cell accumulation in aged (40 days) *Drosophila* midguts (Figs 1B and EV1B). We tested three concentrations (0.01, 0.05, and 0.5 mM) of ALA administration and found 0.5 mM ALA administration showed the best effect of preventing *Esg$^+$* cell accumulation in aged midguts (Fig 1B). Moreover, the luciferase activity of aged flies (40 days) in response to ALA administration started at day 26 after fly eclosion and was even less than that of 26-day flies (intermediate-age flies). Since it has been reported that the ALA level significantly decreases with age in humans (Park *et al*, 2014), this study focused on exploring the role of ALA in preventing stem cell aging.

To further analyze the anti-ISC-aging effect of ALA, ISCs and their differentiating progenies were visualized in aged midguts using an *esg*-GFP reporter line. Simultaneously, the number of ISCs was analyzed, which were identified by the Dl antibody staining and the proliferation rate of ISCs, as indicated by the phosphorylated histone 3 (pH3$^+$; a mark of mitosis) antibody staining. Consistent with previous studies (Biteau *et al*, 2008; Choi *et al*, 2008), the numbers of *esg*-GFP$^+$ cells, Dl$^+$ cells, and pH3$^+$ cells continuously increased with age in the *Drosophila* midguts (Fig 1C–E, G and H). However, 40-day flies (aged flies) that were fed with 0.5 mM ALA for 14 days (starting at the 26$^{th}$ day after eclosion) showed significantly less accumulation of *esg*-GFP$^+$ cells, Dl$^+$ cells, and pH3$^+$ cells compared with 40-day flies without ALA feeding (Figs 1E–H and EV1C). This indicates that ALA can alleviate ISC aging in old flies. More importantly, *esg*-GFP$^+$ cells, Dl$^+$ cells, and pH3$^+$ cells of 40-day flies, that received ALA administration starting at the 26$^{th}$ day, were even less than the cells of 26-day flies (intermediate-age flies; Fig 1D and F–H). This was consistent with the result obtained from the "*esg*-luciferase" reporter system (Fig 1B). This suggested that ALA could not only reduce but also reverse ISC aging based on the change of ISC proliferation rate. In addition, 40-day flies with lifelong ALA administration (starting after fly eclosion) showed less ISC accumulation in midguts. This was found by comparing the numbers of *esg*-GFP$^+$

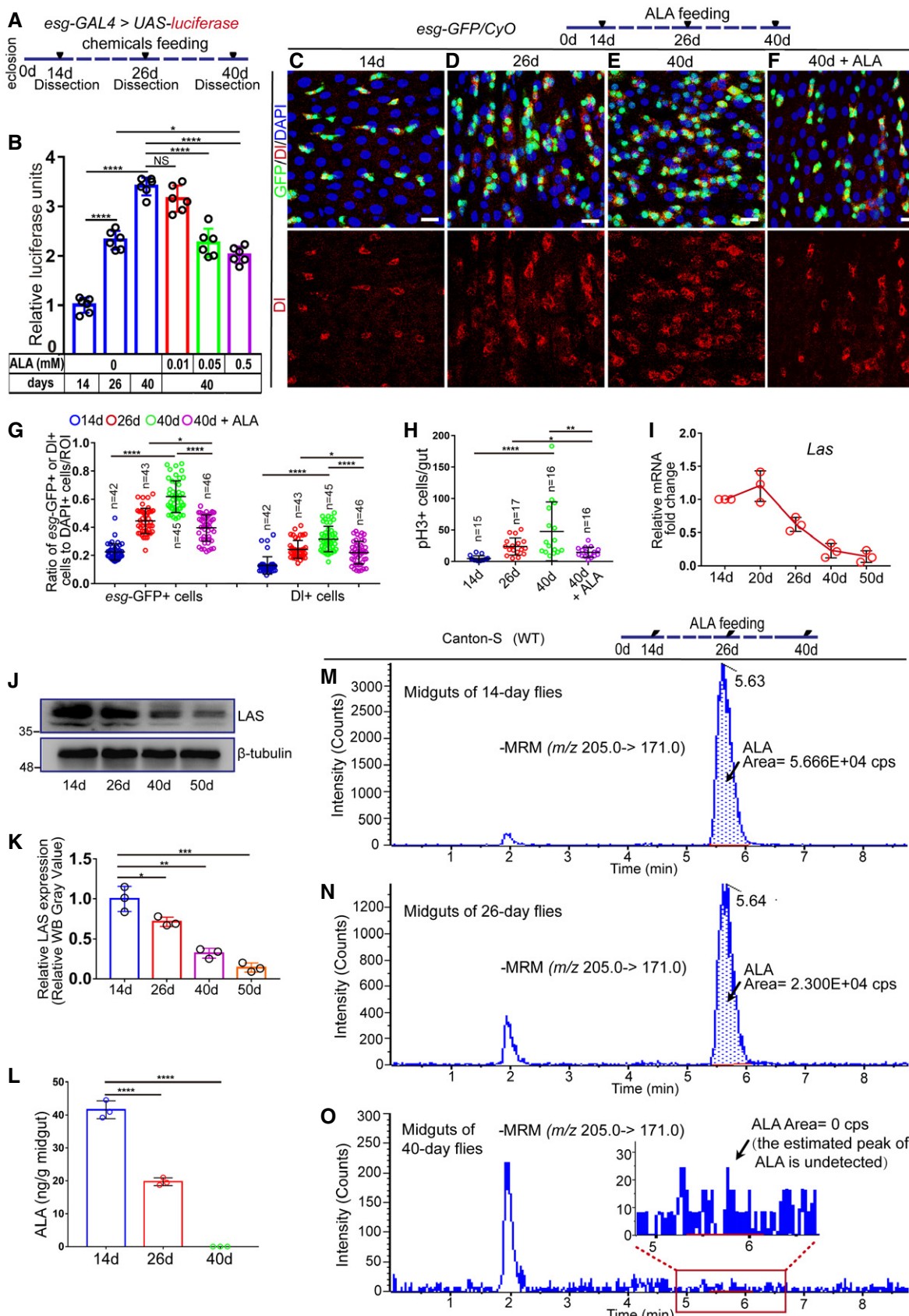

Figure 1.

**Figure 1. Alpha-lipoic acid (ALA) synthesis reduces in aged *Drosophila* midguts, and orally administered ALA rejuvenates aged intestinal stem cells (ISCs).**

A   A model illustrating the *Drosophila* "*esg*>luciferase" reporter system of chemical administration. The chemicals were fed to *Drosophila* with *esg-GAL4*-driven luciferase expression on the 26th day after eclosion. After administration for 14 days, *Drosophila* were dissected, and the activity of luciferase in their midguts was measured.

B   Quantification of the luciferase activity of flies with indicated ages and ALA administration. Error bars show the standard deviation (SD) of six independent experiments.

C–F   Immunofluorescence images of *esg*-GFP and Delta (Dl) staining with the midgut section from the R4 region of 14-day *Drosophila* (C), 26-day *Drosophila* (D), 40-day *Drosophila* (E), and 40-day *Drosophila* in response to 0.5 mM ALA administration, which started at day 26 after eclosion (F). *esg*-GFP (green) identifies ISCs and their differentiating cells. Dl staining (red) was used to visualize ISCs.

G   Quantification of the number of *esg*-GFP$^+$ cells and Dl$^+$ cells in experiments (C–F). *n* is indicated. The numbers of quantified guts from left to right are 15, 17, 16, 16, 15, 17, 16, and 16.

H   Quantification of the number of pH3$^+$ cells in experiments (C-F). *n* is indicated.

I   Relative mRNA fold change of *Las* in sorted *esg*-GFP$^+$ cells of wild-type (*esg-GFP/CyO*) *Drosophila* during aging. The *Las* expressions of *Drosophila* with different ages are plotted relative to 14-day *Drosophila*, which was set to 1. Error bars indicate the SD of three independent experiments.

J   Western blotting results of midguts indicate a decrease of LAS in the midguts of aged *Drosophila*. Loading controls, β-tubulin.

K   Quantification of LAS band intensity as seen in experiments (J). Error bars indicate the SD of three independent experiments.

L   Quantification of the content of ALA in midguts of 14-day, 26-day, and 40-day flies using LC-ESI-MS/MS. Error bars indicate the SD of three independent experiments.

M–O   LC-ESI-MS/MS chromatograms of ALA in midguts of 14-day flies (M), 26-day flies (N), and 40-day flies (O). Arrows indicate the peak of ALA. The area of ALA peak indicates the content of ALA. cps, counts per second; MRM, Multiple Reaction Monitoring. When the estimated peak of ALA is undetected in the chromatogram, the content of ALA is considered as 0.

Data information: DAPI-stained nuclei are shown in blue. Scale bars represent 10 μm (C–F). Error bars represent SDs. Student's *t*-tests, *$P < 0.05$, **$P < 0.01$, ***$P < 0.001$, ****$P < 0.0001$, and non-significant (NS) represents $P > 0.05$. See also Fig EV1.

Source data are available online for this figure.

---

cells, Dl$^+$ cells, and pH3$^+$ cells with those of 40-day flies with ALA administration from an intermediate age (26th day after eclosion; Figs EV1D and 1F). These results show that ALA administration could rejuvenate aged ISCs in *Drosophila*.

**Reduced ALA synthesis in aged flies may cause the functional decline in ISCs**

In animals, ALA is *de novo* synthesized from octanoic acid in mitochondria. ALA synthase (LAS; an iron–sulfur cluster mitochondrial enzyme that catalyzes the final step in the *de novo* pathway of ALA biosynthesis) controls the rate of ALA production in the body (Cronan, 2016; Shaygannia *et al*, 2018; Solmonson & DeBerardinis, 2018). There is only one *Las* homologous gene in *Homo sapiens* and *Drosophila melanogaster*, and the protein sequence of LAS shares 93% similarity and 67% identity. Previous studies have reported a strong inverse correlation between LAS reduction and the status of a number of diseases, including diabetes, atherosclerosis, and neonatal-onset epilepsy (Mayr *et al*, 2011; Padmalayam, 2012; Yi *et al*,

2012; Xu *et al*, 2016). However, whether LAS expression is also decreased during aging remains unknown. To analyze the expression pattern of *Las* in *Drosophila* ISCs during aging, real-time quantitative PCR (RT–qPCR) analyses were performed using sorted *esg*-GFP$^+$ cells (Fig EV1E). These data showed a significant decrease of *Las* transcription in ISCs and their differentiating progenies during aging (Fig 1I). Moreover, Western blotting analysis showed a remarkable decrease of LAS protein in aged midguts of *Drosophila* (Fig 1J–K). To further demonstrate that the abundance of ALA decreases in guts of aged flies, liquid chromatography-electrospray ionization-mass spectrometry (LC-ESI-MS/MS) analyses were performed. These data indicated that the abundance of ALA indeed dramatically decreased in aged midguts of *Drosophila* (Figs 1L–O and EV2A–E). Thus, both the mRNA and protein levels of *Las* in *Drosophila* ISCs undergo a significant reduction in response to aging, which in turn causes a reduction of ALA in midguts of aged flies.

To test whether the reduction of LAS expression in aged ISCs contributes to the age-associated ISC hyperproliferation of old

---

**Figure 2. Reduced ALA synthesis in flies causes the functional decline in ISCs.**

A, B   LC-ESI-MS/MS chromatograms of midguts of flies carrying *Act5C$^{ts}$*-GAL4-driven *UAS-lacZ* (control, A) or *Las RNAi* (B).

C   Quantification of the content of ALA in midguts of flies with indicated genotypes using LC-ESI-MS/MS. Error bars indicate the SD of three independent experiments.

D–F   Immunofluorescence images of the midgut section from the R4 region in *Drosophila* carrying *esg$^{ts}$*-GAL4-driven *UAS-lacZ* (D, control), *Las RNAi* (E), or *Las RNAi* in response to ALA administration (F). *esg*-GFP (green) indicates ISCs and their differentiating cells. Dl staining (red) was used to visualize ISCs.

G   Quantification of the number of *esg*-GFP$^+$ cells, Dl$^+$ cells, and pH3$^+$ cells in experiments (D–F). *n* is indicated. The numbers of quantified guts from left to right are 17, 19, 18, 17, 19, 18, 17, 19, and 18.

H   Quantification of the luciferase activity of midguts with indicated genotypes and manipulation. Error bars show the SD of six independent experiments.

I, J   Immunofluorescence images of control (FRT40A, I) and *Las RNAi* (J) MARCM clones (green, outlined by white dotted lines) 10 days after clone induction (ACI). Pdm1 staining (red) was used to visualize ECs.

K   Quantification of the ratio of Pdm1$^+$ cells per clone of MARCM clones with indicated genotypes. *n* is indicated. Each dot corresponds to one clone.

L   Quantification of MARCM clone size of experiments in (I, J). *n* is indicated. Each dot corresponds to one clone.

Data information: DAPI-stained nuclei are shown in blue. Scale bars represent 10 μm (D–F, I, J). Error bars represent SDs. Student's *t*-tests, ****$P < 0.0001$, and non-significant (NS) represents $P > 0.05$. See also Fig EV2.

Source data are available online for this figure.

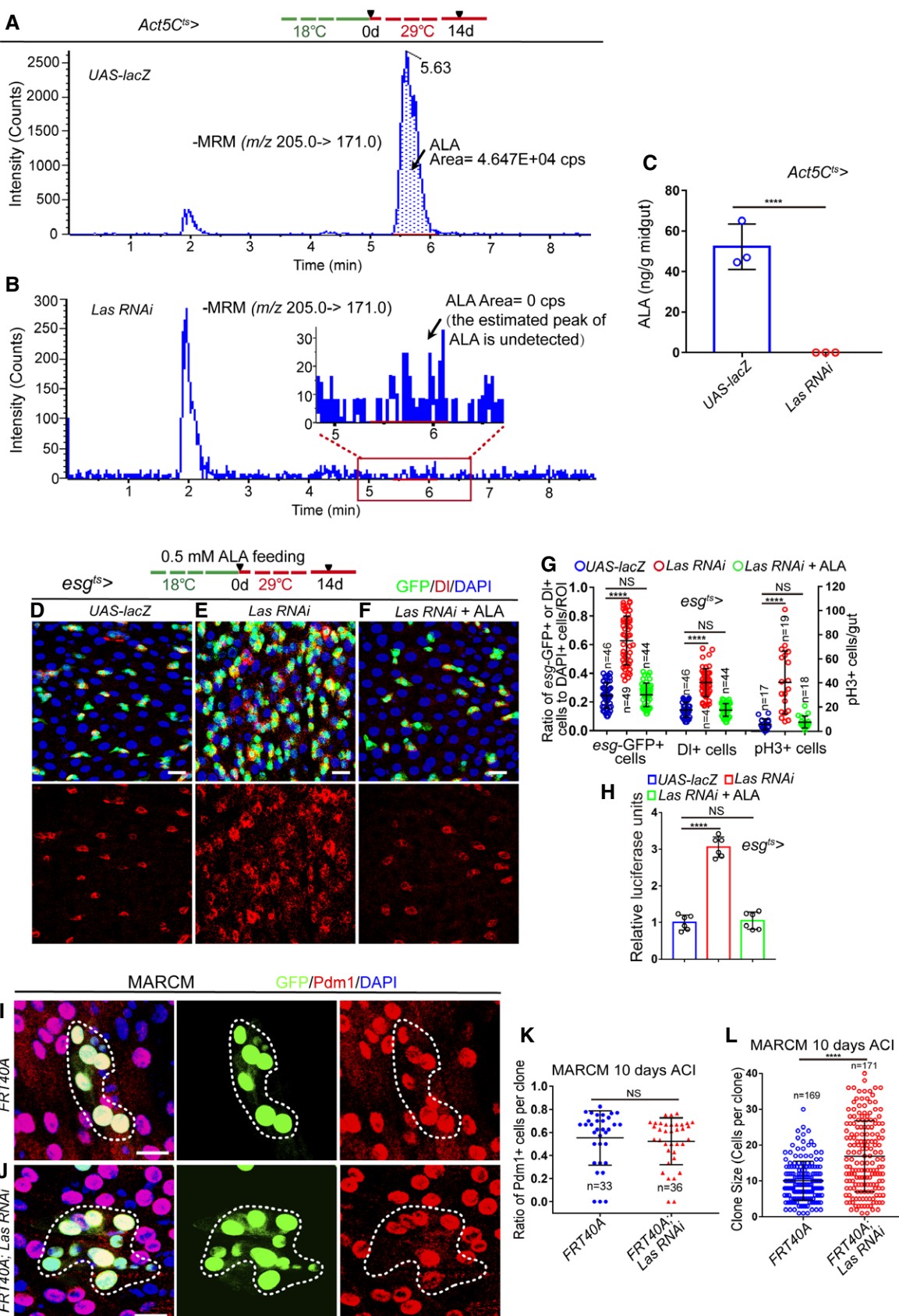

**Figure 2.**

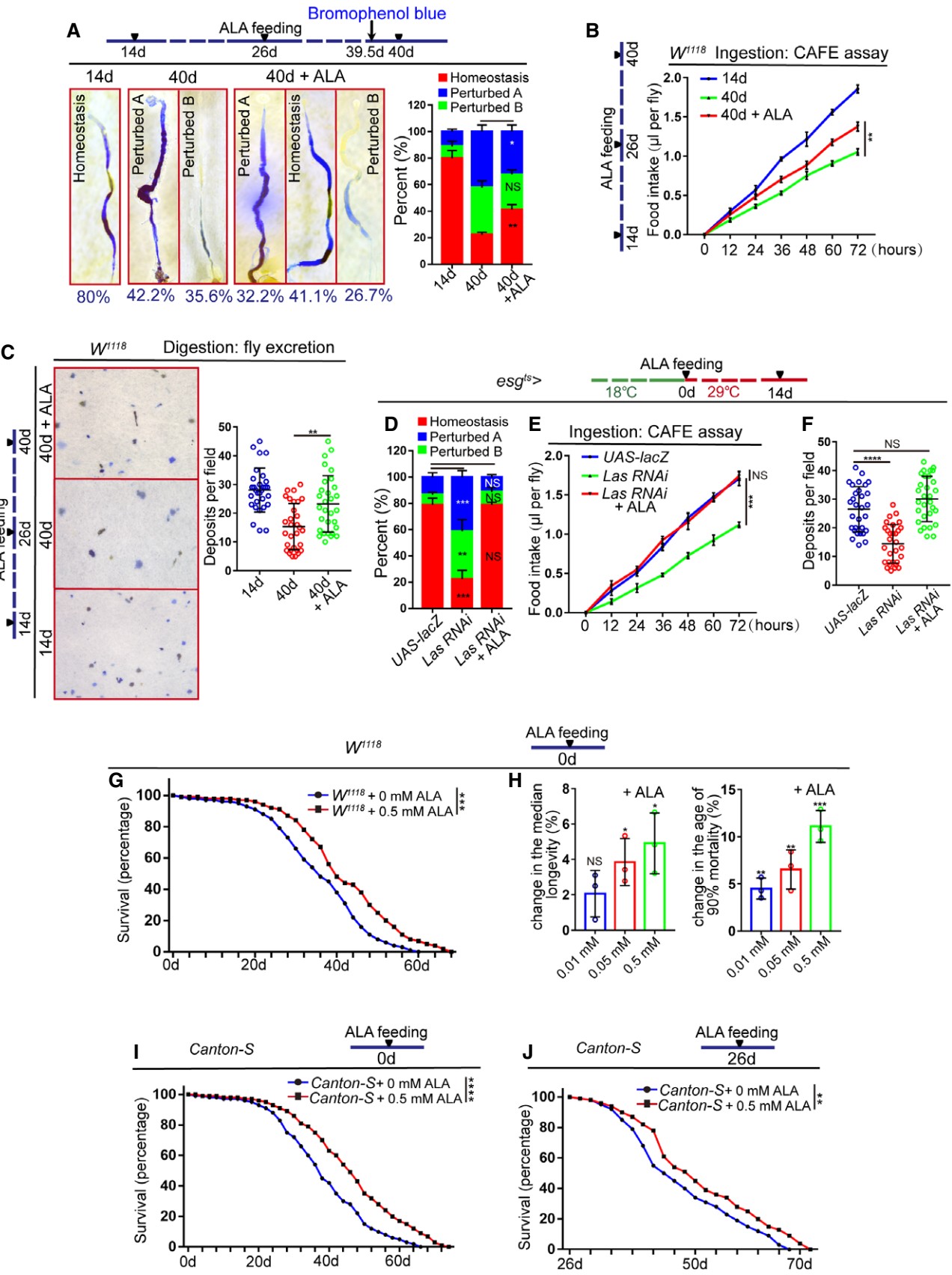

**Figure 3.**

**Figure 3.  ALA prevents the age-related decline of intestinal functions and extends the lifespan of *Drosophila*.**

A   Representative images and quantification of *Drosophila* midguts treated with the pH indicator Bromophenol blue. Moreover, quantification of the three categories of *Drosophila* of indicated conditions. The three categories can be divided into: homeostasis, a well-defined acidic (yellow colored) copper cell region (CCR) is flanked by basic (blue colored) anterior midgut (AM) and posterior midgut (PM). "Perturbed A," the acidic region is lost and the whole gut is basic; "Perturbed B," the strongly acidic region is lost, and the remainder of the gut also becomes less basic. The numbers of quantified guts from left to right are 90, 90, and 90. Error bars show the SD of three independent experiments.

B   Food intake measured using the CAFE assay of *Drosophila* with indicated manipulations. Error bars show the SD of three independent experiments.

C   Excretion of *Drosophila* treated with Bromophenol blue. Images of deposits and quantification of deposit numbers are shown. Excretions are quantified in 30 fields for each group of 12 *Drosophila*. Tests were repeated as three independent experiments.

D   Quantification of three categories of midguts treated with the pH indicator Bromophenol blue in *Drosophila* carrying *esg*[ts]*-GAL4*-driven *UAS-lacZ* (control), *Las RNAi*, or *Las RNAi* with ALA administration. The numbers of quantified guts from left to right are 90, 90, and 90. Error bars show SD of three independent experiments.

E   Food intake measured using CAFE assay of *Drosophila* carrying *esg*[ts]*-GAL4*-driven *UAS-lacZ* (control), *Las RNAi*, or *Las RNAi* with ALA administration. Error bars show the SD of three independent experiments.

F   Quantification of excretion numbers of *Drosophila* carrying *esg*[ts]*-GAL4*-driven *UAS-lacZ* (control), *Las RNAi*, or *Las RNAi* with ALA administration. Excretions are quantified in 30 fields for each group of 12 *Drosophila*. Tests were repeated as three independent experiments.

G   Survival (percentage) of female *W*[1118] *Drosophila* with and without supplementation of ALA as indicated. The numbers of quantified flies: 100 (*W*[1118] + 0.5 mM ALA) and 100 (*W*[1118] + 0 mM ALA). Three independent experiments were conducted.

H   Females median lifespan and females age of 90% mortality of female *W*[1118] *Drosophila* with and without supplementation of ALA as indicated. Error bars show the SD of three independent experiments.

I   Survival (percentage) of female *Canton-S Drosophila* with and without supplementation of ALA as indicated. The numbers of quantified flies: 100 (*Canton-S* + 0 mM ALA), 100 (*Canton-S* + 0.5 mM ALA). Three independent experiments were conducted.

J   Survival (percentage) of female *Canton-S Drosophila* with and without supplementation of ALA at 26-day as indicated. The numbers of quantified flies: 100 (*Canton-S* + 0 mM ALA) and 100 (*Canton-S* + 0.5 mM ALA). Three independent experiments were conducted.

Data information: Error bars represent SDs. *P*-values for lifespan curves (G, I, and J) were calculated by the log-rank test. The statistical tests used in other panels were Student's *t*-tests. *$P < 0.05$, **$P < 0.01$, ***$P < 0.001$, ****$P < 0.0001$, and non-significant (NS) represents $P > 0.05$. See also Fig EV2.

Source data are available online for this figure.

---

midguts, the expression of LAS in ISCs was knocked down using *esg*[ts]*-GAL4*-mediated RNA interference (RNAi; Fig EV2F). LC-ESI-MS/MS analyses indicated that knockdown of LAS led to a significant reduction of ALA synthesis in midguts (Fig 2A–C). Moreover, reduction of LAS expression in young flies indeed caused ISC accumulation (Figs 2D, E and G and H), which was observed in old wild-type *Drosophila* (Fig 1E). More importantly, ALA administration fully reversed the ISC accumulation phenotype caused by LAS depletion in young flies (Fig 2E–H). To further demonstrate that LAS reduction could lead to age-associated ISC hyperproliferation, the lineage of LAS-depleted ISCs was traced by performing mosaic analysis with repressible cell marker (MARCM), which labels all progenies of a single activated ISC in one clone with a visible GFP marker (Micchelli & Perrimon, 2006; Ohlstein & Spradling, 2006).

Based on Pdm1 staining (which labels differentiated ECs), depletion of LAS did not show obvious ISC differentiation defects (Fig 2I–K). However, we found that the average size of LAS-depleted MARCM clones was obviously bigger compared to the control clones (Fig 2I, J and L). This suggested that LAS regulates the rate of ISC proliferation. These findings suggested that the reduced LAS expression in ISCs in response to aging may be a cause of age-related functional decline in ISCs.

## ALA prevents the age-related decline of intestinal functions and extends the lifespan of *Drosophila*

Previous studies have shown that the functional decline of ISCs during aging caused a significant decrease in digestive functions of

---

**Figure 4.  ALA prevents ISC senescence not mainly through its antioxidative ability.**

A–F   Immunofluorescence images of midgut sections from the R4 region in *Drosophila* carrying *esg*[ts]*-GAL4*-driven expression of CAT cDNA (A), CAT cDNA with AD4 administration (B), CAT cDNA with ALA administration (C), *Keap1* RNAi (D), *Keap1* RNAi with AD4 administration (E), or *Keap1* RNAi with ALA administration (F). *esg*-GFP (green) represents ISCs and their differentiating cells. Dl staining (red) was used to visualize ISCs.

G   Quantification of the number of *esg*-GFP+ cells, Dl+ cells of *Drosophila* carrying *esg*[ts]*-GAL4*-driven expression of lacZ cDNA, CAT cDNA, CAT cDNA with AD4 administration, CAT cDNA with ALA administration, *Keap1* RNAi, *Keap1* RNAi with AD4 administration, or *Keap1* RNAi with ALA administration. *n* is indicated. The numbers of quantified guts from left to right are 18, 19, 20, 18, 18, 19, 19, 18, 19, 20, 18, 18, 19, and 19.

H   Quantification of the number of pH3+ cells and luciferase activity of *Drosophila* carrying *esg*[ts]*-GAL4*-driven expression of lacZ cDNA, CAT cDNA, CAT cDNA with AD4 administration, CAT cDNA with ALA administration, *Keap1* RNAi, *Keap1* RNAi with AD4 administration, or *Keap1* RNAi with ALA administration. The numbers of quantified guts from left to right are 18, 19, 20, 18, 18, 19, and 19.

I–L   Representative images of the midgut section from the R4 region of *Drosophila* carrying *esg*[ts]*-GAL4*-driven expression of *lacZ* cDNA (I), *Las RNAi* (J), *Las RNAi* and CAT cDNA (K), or *Las RNAi* and *Keap1* RNAi (L). *esg*-GFP (green) indicates ISCs and their differentiating cells. Dl staining (red) was used to visualize ISCs.

M   Quantification of the number of *esg*-GFP+ cells, Dl+ cells, and pH3+ cells in experiments (I-L). *n* is indicated. The numbers of quantified guts from left to right are 15, 17, 18, 17, 15, 17, 18, 17, 15, 17, 18, and 17.

N   Quantification of luciferase activity of midguts with indicated genotypes. Error bars show the SD of six independent experiments.

Data information: DAPI-stained nuclei are shown in blue. Scale bars represent 10 μm (A–F, and I–L). Error bars represent SDs. Student's *t*-tests, *$P < 0.05$, **$P < 0.01$, ***$P < 0.001$, ****$P < 0.0001$, and non-significant (NS) represents $P > 0.05$.

Source data are available online for this figure.

---

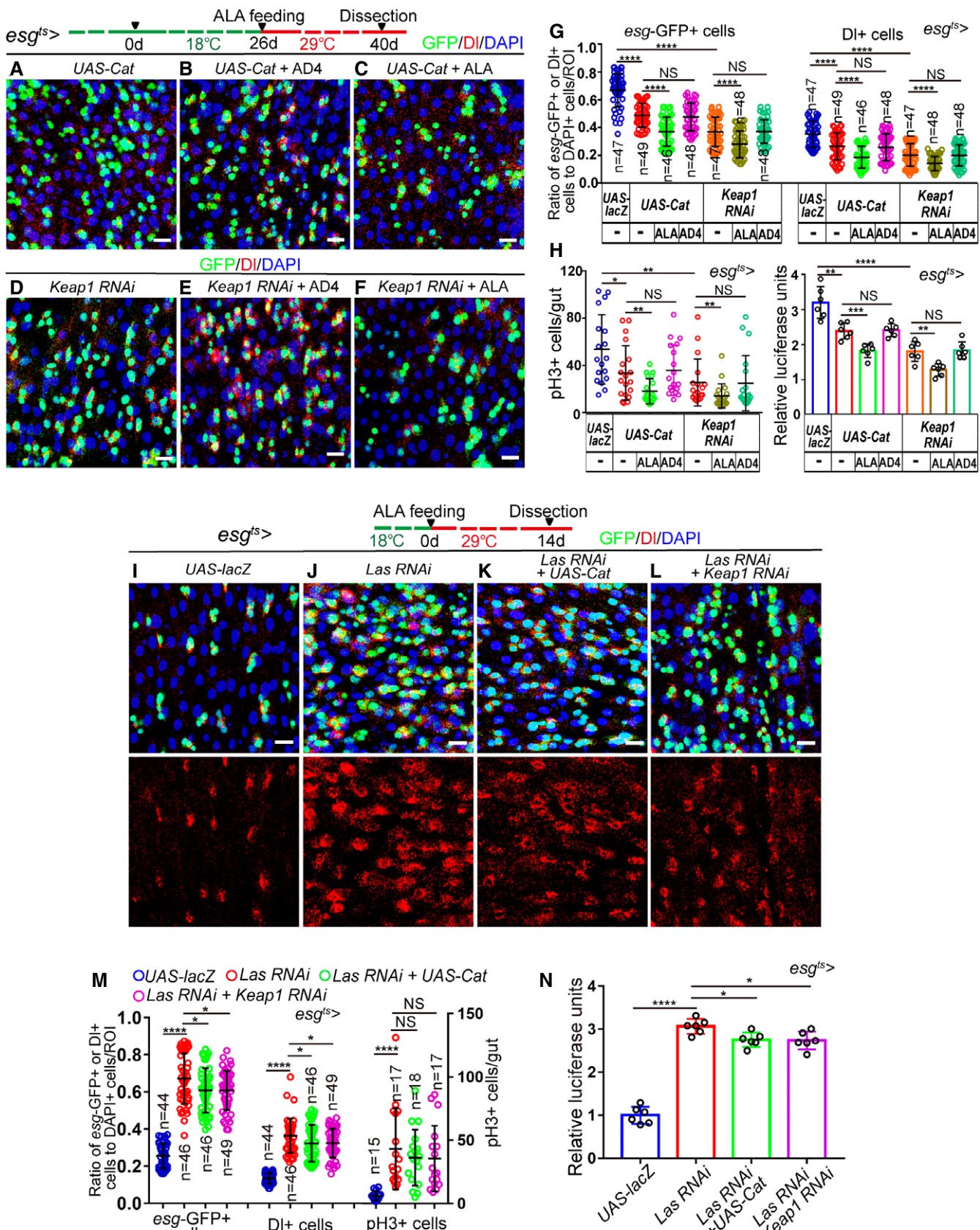

**Figure 4.**

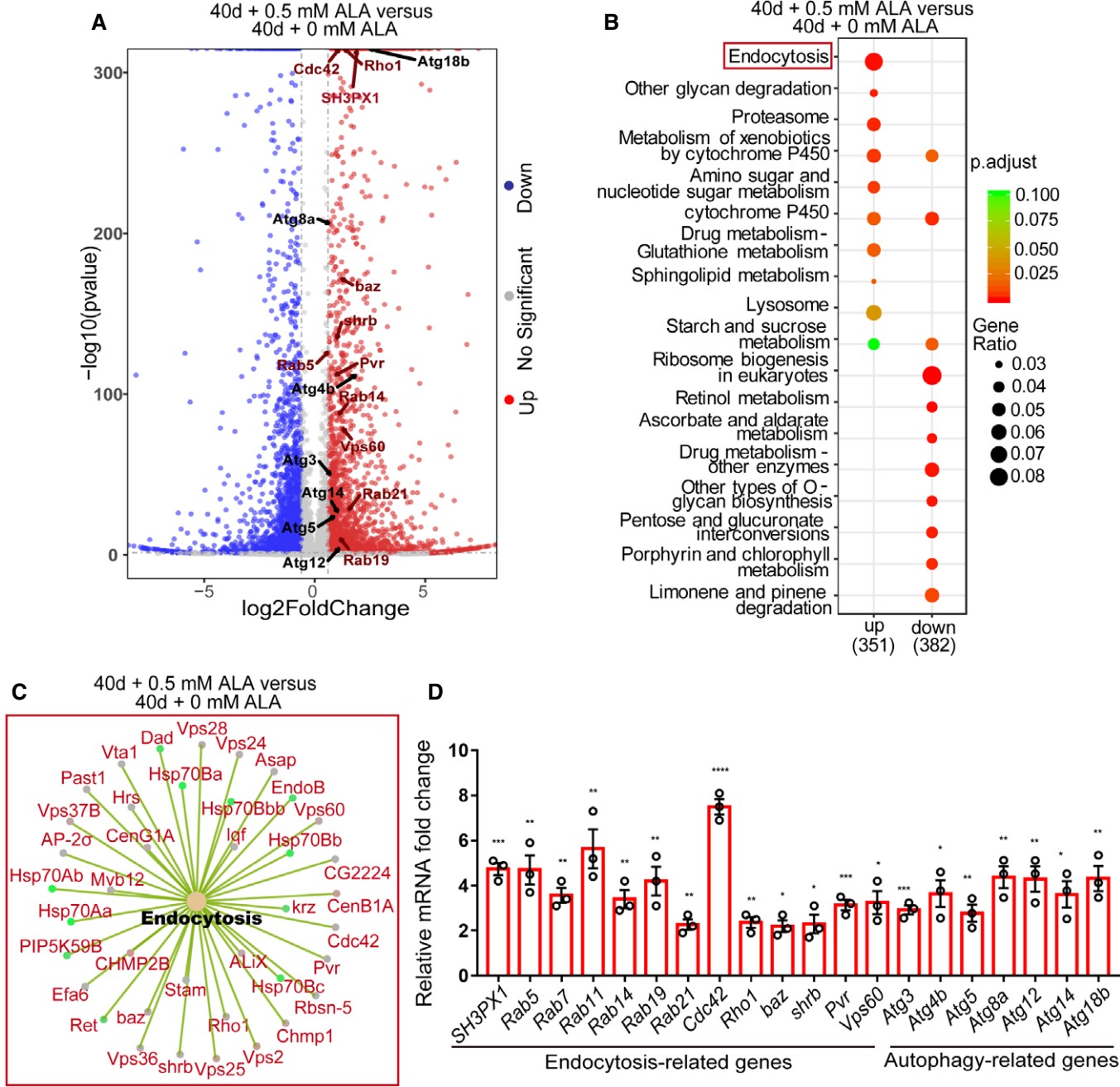

**Figure 5. ALA promotes the upregulation of endocytosis- and autophagy-related genes.**

A  Volcano plots of differentially expressed genes in pair-wise comparison of 40-day *Drosophila* treated without ALA administration to 40-day *Drosophila* treated with 0.5 mM ALA administration. Blue symbols indicate significantly downregulated mass bins (Log$_2$ FC < −0.6 and $P$ < 0.05), red symbols indicate significantly upregulated mass bins (Log$_2$ FC > 0.6 and $P$ < 0.05), and gray symbols indicate mass bins that were not significantly changed.

B  KEGG pathway enrichment analysis of up- or downregulated genes in pair-wise comparison of 40-day *Drosophila* treated without ALA administration to 40-day *Drosophila* treated with 0.5 mM ALA administration. Both adjusted $P$-value and gene ratio denote the significance of the respective pathway.

C  Gene list of endocytosis from KEEG analysis in a pair-wise comparison of 40-day *Drosophila* treated without ALA administration to 40-day *Drosophila* treated with 0.5 mM ALA administration.

D  Relative mRNA fold changes of endocytosis-related and autophagy-related genes with sorted *esg*-GFP⁺ cells of 40-day *Drosophila* treated without ALA administration and 40-day *Drosophila* treated with 0.5 mM ALA administration. Error bars indicate the SD of three independent experiments.

Data information: Error bars represent SDs. Student's $t$-tests, *$P$ < 0.05, **$P$ < 0.01, ***$P$ < 0.001, ****$P$ < 0.0001, and non-significant (NS) represents $P$ > 0.05. See also Fig EV3 and Dataset EV1 and EV2.

Source data are available online for this figure.

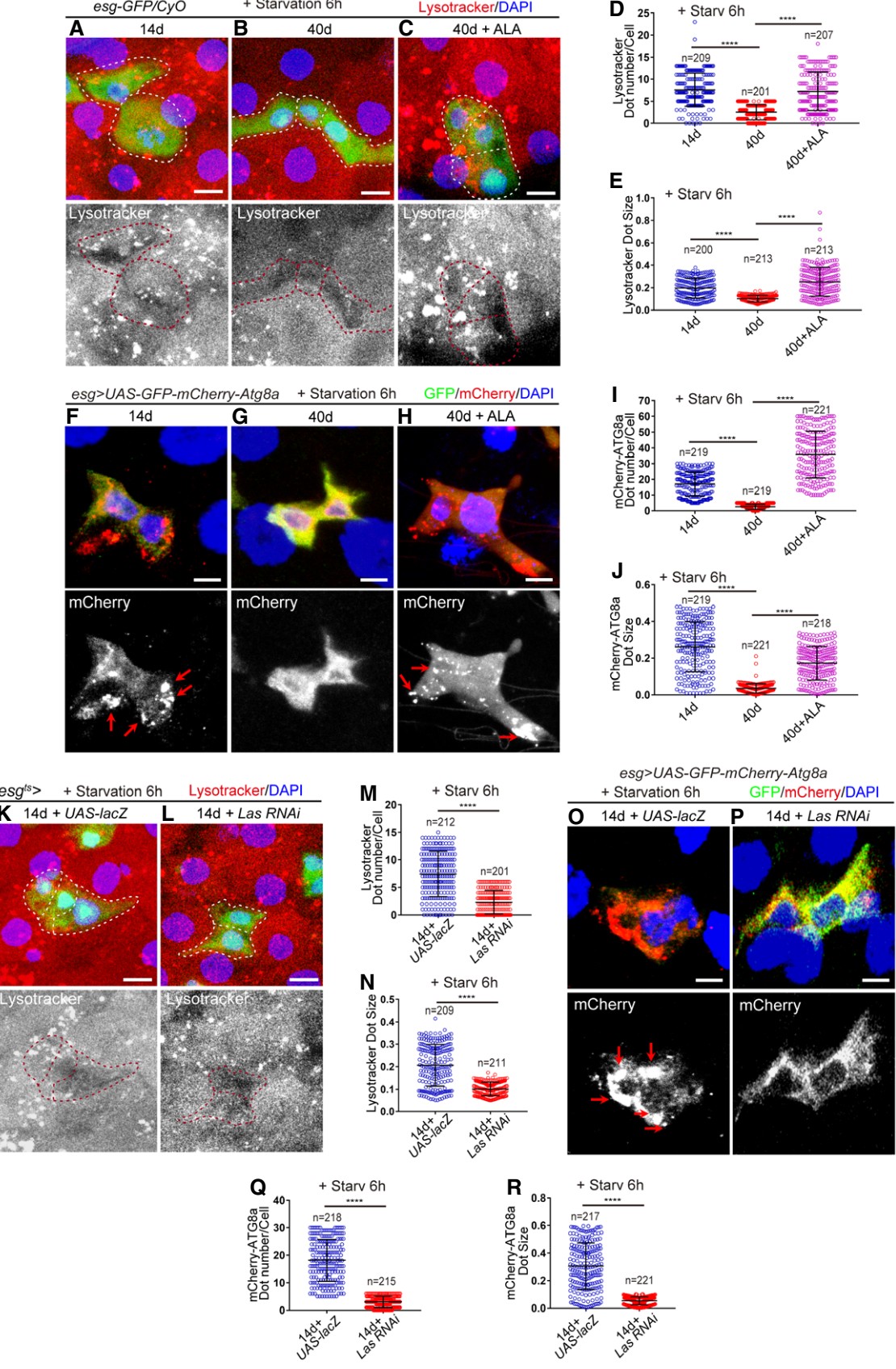

**Figure 6.**

**Figure 6. ALA regulates autophagic activation.**

A–C   Immunofluorescence images of *esg*-GFP and Lysotracker staining with the midgut section from the R4 region in 14-day WT flies (A), 40-day WT flies (B), 40-day WT flies with ALA administration (C). *esg*-GFP (green; outlined by dotted lines), lysotracker (red). *esg*-GFP$^+$ cells are outlined by red dotted lines in the images of lysotracker staining channel.

D   Quantification of the dot number of Lysotracker in *esg*-positive cells from experiments (A-C). *n* is indicated. The numbers of quantified guts from left to right are 9, 11, and 8.

E   Quantification of the dot size of Lysotracker in *esg*-positive cells from experiments (A–C). *n* is as indicated.

F–H   Expression of *esg*-GAL4-driven *UAS-GFP-mCherry-Atg8a* in 14-day *Drosophila* (F), 40-day *Drosophila* (G), 40-day *Drosophila* with ALA administration started at the middle age (26 days) (H). GFP (green) and mCherry (red). The red arrows indicate the autophagosomes.

I   Quantification of the dot number of mCherry in *esg*-positive cells from experiments (F–H). *n* is indicated. The numbers of quantified guts from left to right are 11, 10, and 12.

J   Quantification of the dot size of mCherry in *esg*-positive cells from experiments (F–H). *n* is as indicated.

K, L   Immunofluorescence images of *esg*-GFP and Lysotracker staining with the midgut section from the R4 region in 14-day flies carrying *esg*$^{ts}$-GAL4-driven *UAS-lacZ* (K), and 14-day flies carrying *esg*$^{ts}$-GAL4-driven *Las RNAi* (L). *esg*-GFP (green; outlined by dotted lines), lysotracker (red). *esg*-GFP$^+$ cells are outlined by dotted lines.

M   Quantification of the dot number of Lysotracker in *esg*-positive cells from experiments (K, L). *n* is indicated. The numbers of quantified guts from left to right are 10, and 9.

N   Quantification of the dot size of Lysotracker in *esg*-positive cells from experiments (K, L). *n* is as indicated.

O, P   Expression of *esg*-GAL4-driven *UAS-GFP-mCherry-Atg8a* in 14-day *Drosophila* carrying *esg*$^{ts}$-GAL4-driven *UAS-lacZ* (flies were cultured at 18°C and transferred to 29°C after flies eclosion) (O), and 14-day *Drosophila* with *Las* depleted in ISCs and EBs (flies were cultured at 18°C and transferred to 29°C after flies eclosion) (P). GFP (green) and mCherry (red). The red arrows indicate the autophagosomes.

Q   Quantification of the dot number of mCherry in *esg*-positive cells from experiments (O, P). *n* is indicated. The numbers of quantified guts from left to right are 11, and 10.

R   Quantification of the dot size of mCherry in *esg*-positive cells from experiments (O, P). *n* is as indicated.

Data information: DAPI-stained nuclei are shown in blue. Scale bars represent 5 μm (A–C and K–L) and 2 μm (F–H and O, P). Error bars represent SDs. Student's *t*-tests, ****$P < 0.0001$, and non-significant (NS) represents $P > 0.05$. See also Fig EV4.

Source data are available online for this figure.

*Drosophila*, including the loss of gastrointestinal acid–base homeostasis and a decline of both food intake and excretion (Cognigni *et al*, 2011; Deshpande *et al*, 2014; Li *et al*, 2016). Since ALA administration can reverse age-related ISC proliferation and prevent environmental stress-induced midgut hyperplasia, the effects of ALA administration toward improving the digestive functions of aged flies were further tested. ALA administration did not affect the feeding behavior (food intake) of flies (Fig EV2G). As expected, ALA administration starting at an intermediate age significantly prevented the further deterioration of gastrointestinal acid–base homeostasis (Fig 3A) and alleviated the reduction of both food intake (Fig 3B) and excretion (Fig 3C) in aged flies. Furthermore, reduction of LAS expression in young flies indeed caused a decline of intestinal function similar to that seen in wild-type old flies (Fig 3D–F). ALA administration fully rescued this decline of intestinal function caused by LAS depletion in ISCs (Fig 3D–F). These data suggest that ALA could prevent the ISC-aging-induced functional decline of gastrointestinal tracks.

Since ALA administration significantly reversed ISC aging and relieved the age-associated functional decline of midguts, it was investigated whether ALA administration also extends the lifespan of *Drosophila*. In fact, although this investigation did not provide any detail mechanism, a previous study has shown that lifelong ALA administration could promote *Drosophila* longevity (Teran *et al*, 2012). This experiment was repeated, and a similar result was obtained (Figs 3G–I and EV2H and I). More importantly, we found that feeding the mid-aged (26-day old) flies with ALA also improved their lifespan (Figs 3J and EV2J).

### ALA prevents the age-associated functional decline of ISCs not mainly through its antioxidative ability

Since ALA is well known for its antioxidant ability, and since many studies have demonstrated that ALA administration is

beneficial for multiple diseases by scavenging ROS (Rochette *et al*, 2015; Solmonson & DeBerardinis, 2018), the possibility that ALA prevents the age-associated functional decline in ISCs through its antioxidative ability was first investigated. To test whether ALA prevents the functional decline of ISCs upon aging through its antioxidative ability, the ability of ROS elimination of *Drosophila* ISCs was either reinforced by overexpression of *Catalase* (CAT, the main enzymes of the antioxidant defense system of cells, which catalyze the decomposition of hydrogen peroxide to water and molecular oxygen) or the depletion of *Keap1* (Kelch-like ECH-associated protein 1, which acts as an adaptor of a Cul3 ubiquitin ligase complex). Both of these were reported to fully counteract the increase of ROS in ISCs (Hochmuth *et al*, 2011). If a drug that functions completely through its antioxidative ability prevents ISC hyperproliferation and extends the lifespan of *Drosophila*, it should not enhance the effects of *esg*$^{ts}$-GAL4-mediated CAT overexpression or Keap1 depletion. Indeed, administration of *N*-acetylcysteine amide (AD4; a well-known and strong antioxidant that was used as a positive control) did not further reduce the numbers of *esg*-GFP$^+$ cells, Dl$^+$ cells, and pH3$^+$ cells in aged *esg*$^{ts}$-GAL4-driven midguts either with regard to CAT overexpression or Keap1 depletion (Fig 4A, B, D, E, G and H). However, ALA administration further reduced the numbers of *esg*-GFP$^+$ cells, Dl$^+$ cells, and pH3$^+$ cells in aged midguts, the ISCs of which either overexpressed CAT or were Keap1 depleted, to a much lower level (Fig 4A, C, D and F–H). More importantly, forced induction of *CAT* expression or *Keap1* RNAi only rescued a small ratio of the ISC accumulation phenotype of flies with LAS depleted in ISCs. This was indicated by increases of *esg*-GFP$^+$ cells, Dl$^+$ cells, and pH3$^+$ cells in midguts (Fig 4I–N). These results suggested that scavenging ROS only contributes as a secondary effect of ALA-mediated rejuvenation of aged ISCs. There, ALA must use another antioxidant-independent mechanism to reverse ISC aging.

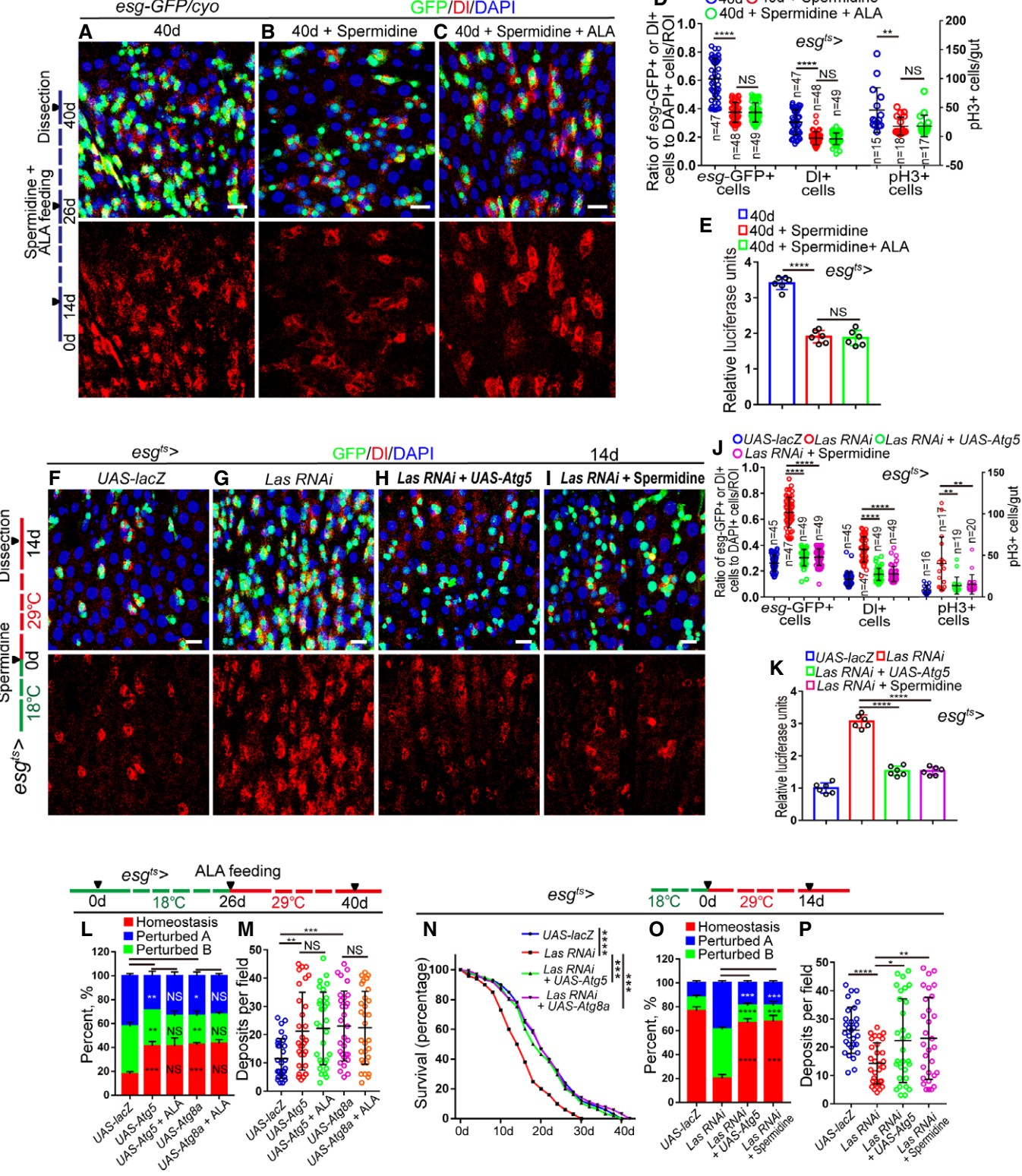

**Figure 7.**

**Figure 7. ALA administration rejuvenates aged ISCs via activation of autophagy process.**

A–C   Immunofluorescence images of midgut section from the R4 region in 40-day flies (A), 40-day flies with spermidine administration started at 26[th] day after fly
        eclosion (B), 40-day flies with spermidine and ALA administration started at 26[th] day after fly eclosion (C). *esg*-GFP (green) indicates ISCs and their differentiating
        cells. Dl (red) staining was used to visualize ISCs.
D     Quantification of the number of *esg*-GFP[+] cells, Dl[+] cells, pH3[+] cells in experiments (A–C). *n* is indicated. The numbers of quantified guts from left to right are: 15,
        18, 17, 15, 18, 17, 15, 18, and 17.
E     Quantification of luciferase activity of midguts indicated genotypes and manipulations. Error bars show the SD of six independent experiments.
F–I    Representative images of the midgut section from the R4 region of *Drosophila* carrying *esg[ts]*-GAL4-driven expression of *lacZ* cDNA (F, control), *Las RNAi* (G), *Las RNAi*
        and Atg5 cDNA (H), or *Las RNAi* with spermidine administration (I). GFP (green) and Dl staining (red) was used to visualize ISCs.
J     Quantification of the number of *esg*-GFP[+] cells, Dl[+] cells, and pH3[+] cells in experiments (F–I). *n* is indicated. The numbers of quantified guts from left to right are
        16, 17, 19, 20, 16, 17, 19, 20, 16, 17, 19, and 20.
K     Quantification of luciferase activity of midguts with indicated genotypes. Error bars show the SD of six independent experiments.
L     Quantification of three categories of midguts treated with the pH indicator Bromophenol blue in *Drosophila* with indicated genotypes. The numbers of quantified
        guts from left to right are 90, 90, 90, 90, and 90. Error bars show the SD of three independent experiments.
M     Quantification of excretion numbers of *Drosophila* with indicated genotypes. Excretions are quantified in 30 fields for each group of 12 *Drosophila*. Tests were
        repeated as three independent experiments.
N     Survival (percentage) of female *Drosophila* with indicated genotypes. The numbers of quantified *Drosophila*: 100 (*UAS-lacZ*), 100 (*Las RNAi*), 100 (*Las RNAi + UAS-
        Atg5*), and 100 (*Las RNAi + UAS-Atg8a*). Three independent experiments were conducted.
O     Quantification of three categories of midguts treated with the pH indicator Bromophenol blue in *Drosophila* with indicated genotypes. The numbers of quantified
        guts from left to right are 90, 90, 90, and 90. Error bars show the SD of three independent experiments.
P     Quantification of excretion numbers of *Drosophila* with indicated genotypes. Excretions are quantified in 30 fields for each group of 12 *Drosophila*. Tests were
        repeated as three independent experiments.

Data information: DAPI-stained nuclei are shown in blue. Scale bars represent 10 μm (A–C and F–I). Error bars represent SDs. *P*-values for lifespan curves (N) were
calculated by the log-rank test. The statistical tests used in other panels were Student's *t*-tests. *$P < 0.05$, **$P < 0.01$, ***$P < 0.001$, ****$P < 0.0001$, and non-significant
(NS) represents $P > 0.05$. See also Figs EV4 and EV5.
Source data are available online for this figure.

## ALA administration promotes the upregulation of endocytosis- and autophagy-related genes in the aged midguts of *Drosophila*

To identify the mechanism with which ALA rejuvenates aged ISCs, RNA-sequencing (RNA-seq) was performed on dissected midguts of *Drosophila* both with and without ALA administration (results listed in Dataset EV1). The results of this study showed that a cluster of autophagy-related genes (such as *Atg3*, *Atg4b*, *Atg5*, *Atg8a*, *Atg12*, *Atg14*, and *Atg18b*) had a significantly higher level in aged midguts (40 days) of *Drosophila* that received ALA administration compared with old midguts of *Drosophila* that did not receive ALA administration (Fig 5A, and Dataset EV2). Moreover, Kyoto Encyclopedia of Genes and Genomes (KEGG) pathway enrichment analysis of differentially expressed genes, which were only upregulated in old midguts of *Drosophila* that received ALA administration, showed significant enrichment of genes (such as *Rab5*, *Rab14*, *Rab19*, *Rab21*, *Rho1*, *Vps60*, *Cdc42*, *Baz*, *Pvr*, and *Shrb*) involved in endocytosis (Figs 5A–C and EV3, and Dataset EV2). Among these upregulated endocytosis genes, *Rab5* (Zhang *et al*, 2019), *Cdc42* (Morin-Poulard *et al*, 2016), *Pvr* (Bond & Foley, 2012; Ferguson & Martinez-Agosto, 2017), and *Vps60* (Berns *et al*, 2014) had been reported to regulate *Drosophila* ISC proliferation under normal conditions.

To confirm the results of these RNA-seq analyses and to eliminate the interference of other midgut cells (ECs, EES, and muscle cells), RT–qPCR analyses were performed using sorted ISCs and EBs (*esg*-GFP[+] cells; Fig EV1E). The results of these RT–qPCR analyses of the selected genes (including several endocytosis and autophagy genes) showed similar expression patterns than RNA-seq analysis (Fig 5D). Furthermore, *Rab7*, a key regulator of late endosome formation, was also significantly upregulated (Fig 5D). These findings strongly suggest that ALA rejuvenates aged ISCs by promoting the expression of specific autophagy- and endocytosis-related genes in aged ISCs.

## ALA administration rejuvenates aged ISCs by activating the autophagy process

Activation of autophagy has long been linked to anti-aging in diverse model systems (Martinez-Lopez *et al*, 2015; Filfan *et al*, 2017; Revuelta & Matheu, 2017); therefore, it was first assessed whether ALA reverses aged ISCs through an autophagy-mediated mechanism. To visualize the autophagy activity in ISCs, lyso-tracker (an acid-tropic dye that can be used to detect autophagy-associated lysosomal activity) and ATG8a-GFP-mCherry reporter (a reporter line that can be used to measure the dynamic autophagic flux (Mauvezin *et al*, 2014)) were used to label ISCs in old midguts both with and without ALA administration. As indicated by lysotracker staining, young ISCs (14 days; Fig 6A, D and E) showed clearly higher autophagy activity than aged ISCs (40 days; Fig 6B, D and E), while ALA administration significantly increased the autophagy activity in aged ISCs (Fig 6C–E). Consistently, ISCs of flies that carry *esg*-GAL4-driven *UAS-GFP-mCherry-Atg8a* showed more and bigger mCherry foci in young ISCs than in old ISCs (Figs 6F, G, I and J, and EV4A and B). Moreover, ALA administration significantly increased the autophagy activity in ISCs of old flies as indicated by the formation of more and larger mCherry foci (Figs 6G–J and EV4B and C). Moreover, disruption of the ALA synthesis in ISCs of young *Drosophila* by *esg[ts]*-GAL4-mediated *Las* RNAi also yielded a significant reduction of the autophagy activity (Figs 6K–R and EV4D and E). Interestingly, we found that depletion of LAS in *esg*[+] cells caused the reduction of lysotracker signal in all midgut epithelial cells. We think the possible reason is that depletion of LAS in *esg*[+] cells led to the whole midgut (which has a high turnover rate and maintains by ISCs) undergo premature aging, which mimicked the old midguts and exhibited a global reduction of autophagy activity in the whole midgut epithelia.

Activation of autophagy in aged *Drosophila* by feeding with spermidine (an chemical autophagy inducer (Eisenberg *et al*, 2009; Sigrist *et al*, 2014)) or by overexpression of ATG5 or ATG8a (two

proteins that had been reported to promote the basal level of autophagy in aged animals and extend lifespan(Simonsen *et al*, 2008; Pyo *et al*, 2013)) significantly inhibited the increase of *esg*-GFP⁺ cells,

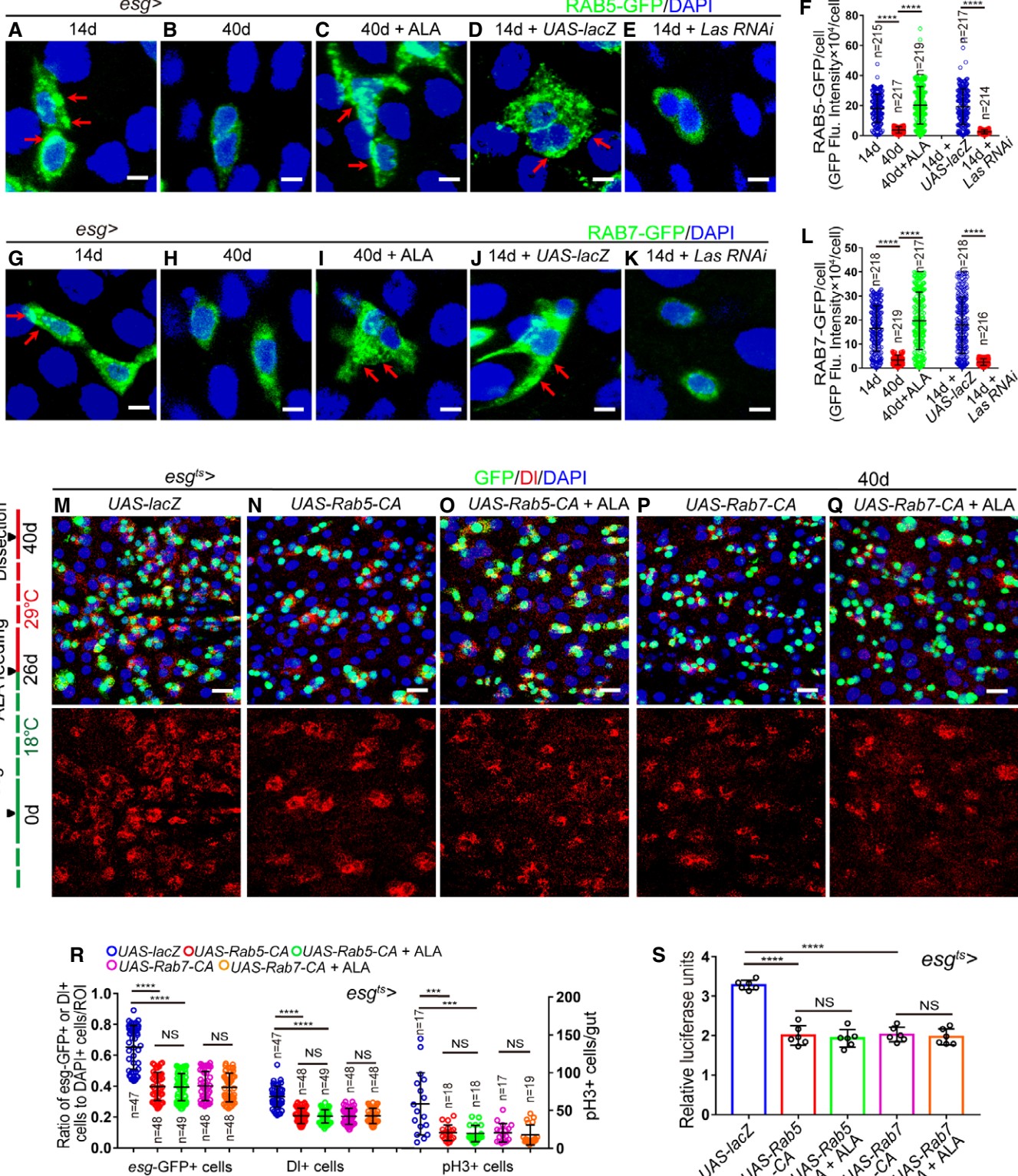

**Figure 8.**

◀

**Figure 8.  ALA administration rejuvenates aged ISCs by modulating endocytosis.**

A–E     Expressions of Rab5-GFP reporter in ISCs of 14-day WT *Drosophila* (A), 40-day WT *Drosophila* (B), 40-day *Drosophila* with ALA administration (C), 14-day *Drosophila* carrying *esg^{ts}-GAL4>UAS-lacZ* (D), and 14-day *Drosophila* carrying *esg^{ts}-GAL4>Las RNAi* (E). GFP (green).
F       Quantification of fluorescence intensity of RAB5-GFP in experiments (A–E). Each dot corresponds to one cell. *n* is indicated. The numbers of quantified guts from left to right are 13, 11, 12, 10 and 13.
G–K     Expressions of Rab7-GFP reporter in ISCs of 14-day WT *Drosophila* (G), 40-day WT *Drosophila* (H), 40-day *Drosophila* with ALA administration (I), 14-day *Drosophila* carrying *esg^{ts}-GAL4>UAS-lacZ* (J), and 14-day *Drosophila* carrying *esg^{ts}-GAL4>Las RNAi* (K). GFP (green).
L       Quantification of fluorescence intensity of RAB7-GFP in experiments (G–K). Each dot corresponds to one cell. *n* is indicated. The numbers of quantified guts from left to right are 11, 10, 11, 13, and 12.
M–Q     Immunofluorescence images of midgut section from the R4 region of *Drosophila* carrying *esg^{ts}-GAL4*-driven expression of *lacZ* cDNA (M, control), constitutively active form of RAB5 (N), constitutively active form of RAB5 with ALA administration (O), constitutively active form of RAB7 (P), or constitutively active form of RAB7 with ALA administration (Q). GFP (green) and Dl staining (red) was used to visualize ISCs.
R       Quantification of the number of *esg*-GFP[+] cells, Dl[+] cells, and pH3[+] cells in experiments (M–Q). *n* is indicated. The numbers of quantified guts from left to right are 17, 18, 18, 17, 19, 17, 18, 18, 17, 19, 17, 18, 18, 17, and 19.
S       Quantification of luciferase activity in *Drosophila* with indicated genotypes and applied manipulations. Error bars show the SD of six independent experiments.

Data information: DAPI-stained nuclei are shown in blue. Scale bars represent 2 μm (A–E, G–K), and 10 μm (M–Q). Error bars represent SDs. Student's *t*-tests, ***$P < 0.001$, ****$P < 0.0001$, and non-significant (NS) represents $P > 0.05$. See also Fig EV5.
Source data are available online for this figure.

Dl[+] cells, and pH3[+] cells in old midguts (Figs 7A, B, D and E, and EV4F and H–K). ALA administration could not further decrease the numbers of *esg*-GFP[+] cells, Dl[+] cells, and pH3[+] cells in *Drosophila* that carry *esg^{ts}-GAL4*-driven ATG5, ATG8a overexpression (Figs 7B–E and EV4G–L), or were fed with spermidine (Fig 7C–E). The phenotype of ISC accumulation (an increase of *esg*-GFP[+] cells, Dl[+] cells, and pH3[+] cells in midguts), caused by LAS depletion in ISCs, was largely rescued by ATG5 overexpression or spermidine administration (Fig 7F–K). However, ALA administration could not decrease the defect of ISC hyperproliferation caused by the depletion of ATG6 or ATG8a in ISCs of young *Drosophila* (Fig EV5A–G). These findings suggested that autophagy functions downstream of ALA and prevents the functional decline of ISCs in response to aging.

Next, the effects of ALA and autophagy on digestive improvement (indicated by gastrointestinal acid–base homeostasis and excretion) of aged flies were investigated. As expected, activation of the autophagy process in aged ISCs via overexpression of ATG5 or ATG8a in ISCs prevented the deterioration of gastrointestinal acid–base homeostasis as well as the decrease of excretion of aged *Drosophila* (Fig 7L and M). ALA administration could not further promote autophagy-mediated digestive improvement (Fig 7L and M). In addition, forced expression of *Atg8a* or *Atg5* cDNA in ISCs significantly rescued the phenotypes of shortened lifespan, deterioration of gastrointestinal acid–base homeostasis, and decreased digestive function that were found in *Drosophila* with LAS depleted in ISCs (Figs 7N–P and EV4M). Consistently, ALA could not ameliorate the phenotypes of gastrointestinal acid–base homeostasis deterioration and declined digestive function that were found in *Drosophila* with ATG6 or ATG8a depleted in ISCs (Fig EV5H and I). These results demonstrated that autophagy functions downstream of ALA and promotes digestive function in aged *Drosophila*.

### ALA administration rejuvenates aged ISCs by modulating endocytosis

A recent study has shown that endocytosis worked upstream of autophagy to regulate the ISC division in *Drosophila* midguts (Zhang *et al*, 2019). Furthermore, the KEGG analysis of this study identified the endocytosis process as the most affected pathway that was upregulated in old midguts with ALA administration (see Fig 5B

and C). Therefore, this study investigated whether ALA rejuvenates aged ISCs and extends the lifespan of *Drosophila* via modulation of the endocytosis process. Consequently, RAB5-labeled early endosomes and RAB7-labeled late endosomes upon aging were first visualized in ISCs and EBs, respectively. Immunofluorescence analyses clearly showed that both RAB5-labeled early endosomes and RAB7-labeled late endosomes significantly decreased in ISCs and EBs of old *Drosophila* (Fig 8A, B, F–H, and L). Moreover, disruption of ALA synthesis led to a remarkable reduction of both RAB5-labeled early endosomes and RAB7-labeled late endosomes (Fig 8D–F and J–L). Interestingly, after ALA supplementation, the abundances of both RAB5-labeled early endosomes and RAB7-labeled late endosomes in ISCs and EBs of old *Drosophila* returned to a similar level of young *Drosophila* (Fig 8C, F, H, I and L). These results suggested that ALA regulates endosome formation and maturation in the ISCs of *Drosophila*.

To investigate whether the endocytosis process participates in preventing ISC aging, the constitutively active versions of either RAB5 (RAB5-CA) (Li & Stahl, 1993; Stenmark *et al*, 1994) or RAB7 (RAB7-CA) (Lorincz *et al*, 2017; Li *et al*, 2018) were forcedly expressed in the ISCs of middle-aged *Drosophila*. Re-activating the endocytosis process in aged ISCs by overexpressing either RAB5-CA or RAB7-CA significantly inhibited the age-associated ISC hyperproliferation in midguts of old *Drosophila* (Fig 8M, N, P, R and S). In addition, ALA administration could not further enhance the anti-ISC-aging effect induced by RAB5-CA or RAB7-CA overexpression in ISCs of old *Drosophila* (Fig 8N–S). Importantly, forced expression of RAB5-CA or RAB7-CA by *esg^{ts}-GAL4* significantly rescued the defect of ISC hyperproliferation observed in *Drosophila* with LAS depletion in ISCs (Fig 9A–F). However, ALA administration could not reduce the ISC hyperproliferation defect caused by the depletions of RAB5 or RAB7 (Figs 9G and H, and EV5J–N). These results indicate that ALA functions upstream of endocytosis to prevent the functional decrease of ISCs in response to aging.

In addition, orally administrated ALA could not further prevent the deterioration of gastrointestinal acid–base homeostasis or enhance the digestive function of the intestine of *Drosophila* with RAB5-CA or RAB7-CA overexpression in ISCs (Fig 10A and B). Forced expressions of either RAB5-CA or RAB7-CA significantly rescued the shortened lifespan, the deterioration of gastrointestinal

acid–base homeostasis, and the declined digestive function observed in *Drosophila* with LAS depleted in ISCs (Figs 10C–E and EV5O). However, ALA administration could not rescue the defects of deterioration of gastrointestinal acid–base homeostasis and the declined

digestive function that observed in flies with RAB5 or RAB7 depleted in ISCs (Fig EV5P and Q). Thus, this study demonstrated that endocytosis works downstream of ALA in ISCs to protect the degeneration of intestinal function upon aging.

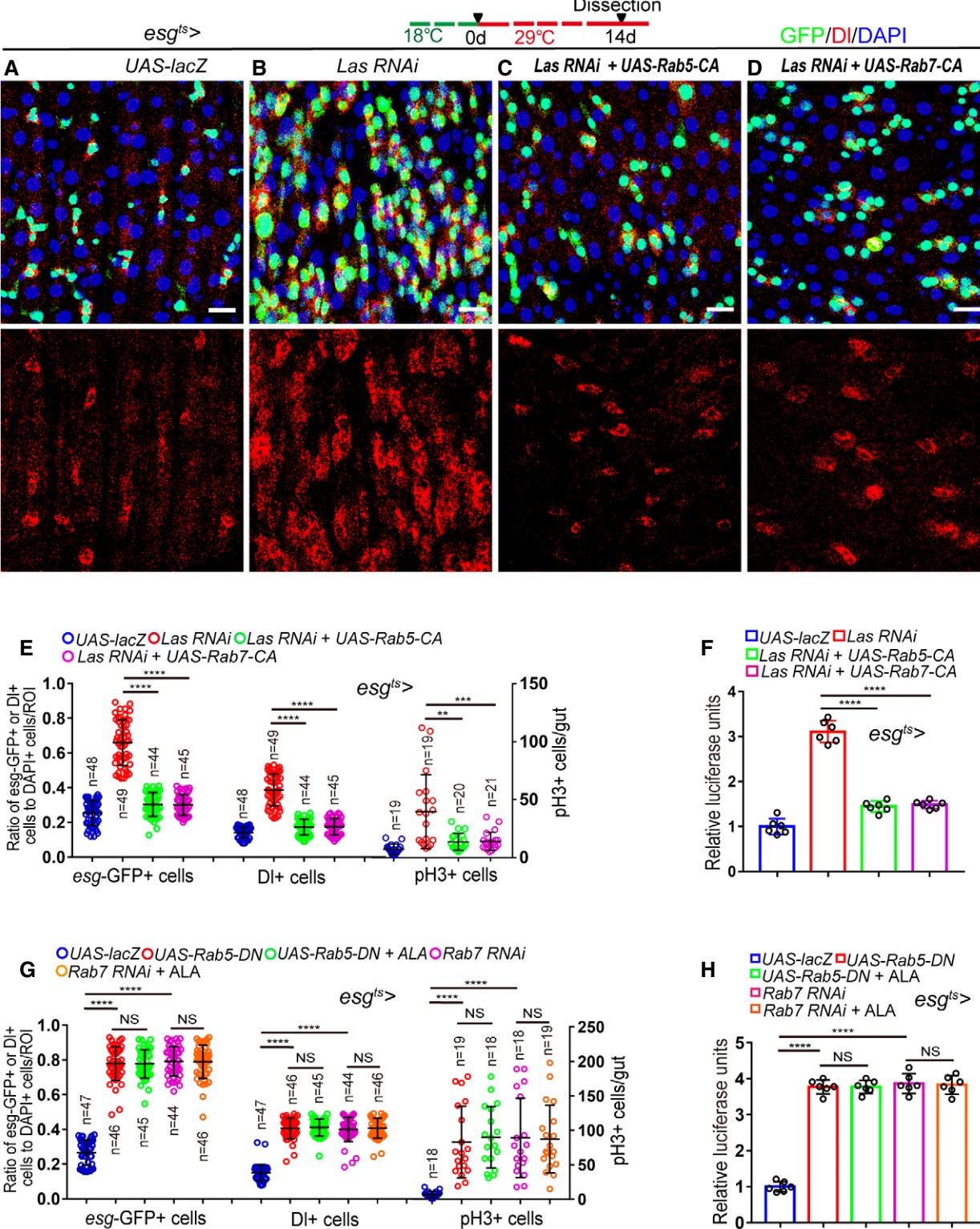

**Figure 9.**

**Figure 9.   ALA functions upstream of endocytosis to prevent the functional decrease of ISCs.**

A–D   Representative images of midgut sections from the R4 region of *Drosophila* carrying *esg*^ts-GAL4-driven expression of *lacZ* cDNA (A, control), *Las RNAi* (B), *Las RNAi* and *RAB5-CA* (C), or *Las RNAi* and *RAB7-CA* (D). GFP (green), Dl staining (red) was used to visualize ISCs.

E       Quantification of the number of *esg*-GFP⁺ cells, Dl⁺ cells, and pH3⁺ cells in experiments (A–D). *n* is indicated. The numbers of quantified guts from left to right are 19, 19, 20, 21, 19, 19, 20, 21, 19, 19, 20, and 21.

F       Quantification of the luciferase activity of midguts of *Drosophila* with indicated genotypes and manipulations. Error bars show the SD of six independent experiments.

G       Quantification of *esg*-GFP⁺ cells, Dl⁺ cells, and pH3⁺ cells in midguts of *Drosophila* with indicated genotypes and manipulations. *n* is indicated. The numbers of quantified guts from left to right are 18, 19, 18, 18, 19, 18, 19, 18, 18, 19, 18, 19, 19, 18, 18, and 19.

H       Quantification of luciferase activity of midguts of *Drosophila* with indicated genotypes and manipulations. Error bars show the SD of six independent experiments.

Data information: DAPI-stained nuclei are shown in blue. Scale bars represent 10 μm (A–D). Error bars represent SDs. Student's *t*-tests, **$P$ < 0.01, ***$P$ < 0.001, ****$P$ < 0.0001, and non-significant (NS) represents $P$ > 0.05. See also Fig EV5.

Source data are available online for this figure.

### ALA restrains ISC hyperproliferation in aged *Drosophila* by counteracting endocytosis-autophagy-mediated EGFR activity

A recent study indicated EGFR signaling as the main trigger for ISC overproliferation when the endocytosis–autophagy network was disturbed (Zhang *et al*, 2019). Therefore, this study investigated the EGFR activity of ISCs of *Drosophila* in response to ALA administration by monitoring the dpERK level. ALA administration considerably reduced the activation of EGFR signaling in ISCs of aged *Drosophila* (Fig 10F–G and J). The depletion of LAS in ISCs caused a significant upregulation of EGFR activity (Fig 10H–J). Importantly, the forced expression of a dominant-negative version of *Egfr* (*Egfr-DN*) significantly rescued the ISC overproliferation observed in LAS-depleted *Drosophila* (Fig 10K–O). Thus, ALA rejuvenates aged ISCs by counteracting endocytosis-autophagy-mediated EGFR activity.

## Discussion

The ubiquitous but often overlooked endogenous small molecule ALA remained almost completely unexplored in the context of stem-cell-mediated tissue degeneration and organismal aging. This study found a significant downregulation of the expression of ALA synthetase (LAS) in aged ISCs. In *Drosophila*, both old wild-type midguts and young midgut with LAS depletion in ISCs showed decreased endocytosis activity, which in turn inhibited the autophagy process and induced the over activation of EGFR signaling. Ultimately, this leads to the hyperproliferation of ISCs with age (Fig 10P). Oral administration of ALA significantly reversed the defects of ISC hyperproliferation and gut functional decline in both aged wild-type and young *Drosophila* with LAS depletion in ISCs.

Over the past decade, studies on ALA were mostly performed with a focus on its anti-oxidation function and the use of ALA for the reduction of blood glucose (Ghelani *et al*, 2017; Akbari *et al*, 2018). However, other functions of ALA, especially in the context of stem cell regulation, have been largely neglected. In fact, several previous studies provided evidence that ALA also participates in the activation of signaling pathways (such as Jak/Stat signaling and TNFα signaling) and the regulation of gene transcription (Choi *et al*, 2016; Wu *et al*, 2016). The present study represents a first step and indicates that ALA plays an important role in the prevention of the functional decline of stem cells in response to aging. More importantly, this study showed that ALA inhibits ISCs overproliferation

not by its ability to eliminate ROS but by promoting the activity of the endocytosis–autophagy network.

Endocytosis refers to the process of active transportation of extracellular material into cells via their membrane. Over the recent decades, abundant research contributed to an increased understanding of the endocytosis process. This process has been shown to regulate multiple important cellular functions, such as nutrient uptake, plasma membrane recycling, antigen presentation, intracellular signal transduction, the removal of aged or dead cells from the body, and the defense against microbes (Doherty & McMahon, 2009; Di Fiore & von Zastrow, 2014). The impairment of the endocytic pathway has been linked to a number of human diseases, such as diabetes, neurodegenerative diseases, cardiovascular disease, and cancer (Ellinger & Pietschmann, 2016). Although the endocytic pathway has been studied in detail in many biological processes and has been demonstrated to exert multiple important cellular functions, its role in the regulation of stem cell functions still remains largely unexplored. A recent study showed that an SH3PX1-dependent endocytosis–autophagy network inhibits ISC proliferation in *Drosophila* midguts by restraining EGFR pathway activity (Zhang *et al*, 2019). This study demonstrated that blockages within the endocytosis–autophagy network stabilize ligand-activated EGFRs. This blocks the recycling of EGFRs to the plasma membrane and, in turn, counteracts EGFR signaling (Zhang *et al*, 2019). However, the biological functions and the physiological significance of the endocytic pathway in the context of ISC aging remain inadequately understood. The current study represents a first step to alleviate this lack of knowledge and indicates that the ALA-regulated endocytic pathway plays an essential role in stem cell regulation and the functional maintenance of tissue upon aging. The presented findings emphasize the need to consider the role of both ALA and endocytosis in the contexts of stem-cell-mediated tissue degeneration and aging.

Although many studies indicated that ALA has multiple cellular functions, ALA was mainly known as an antioxidant and was widely used as a racemic drug to lower the blood sugar levels in patients with diabetes and to alleviate diabetic polyneuropathy-associated pain and paresthesia (Shay *et al*, 2009; Park *et al*, 2014; Salehi *et al*, 2019). The current study shows that ALA can regulate the activity of the endocytic pathway in stem cells. This is the first report that links ALA with the endocytosis–autophagy network and its regulated EGFR signaling. Although the detailed mechanism of how ALA activates the endocytic pathway still remains unexplored, the findings of the present study suggest an interplay between ALA and endocytosis, which restrains the overactivity of EGFR and ISC

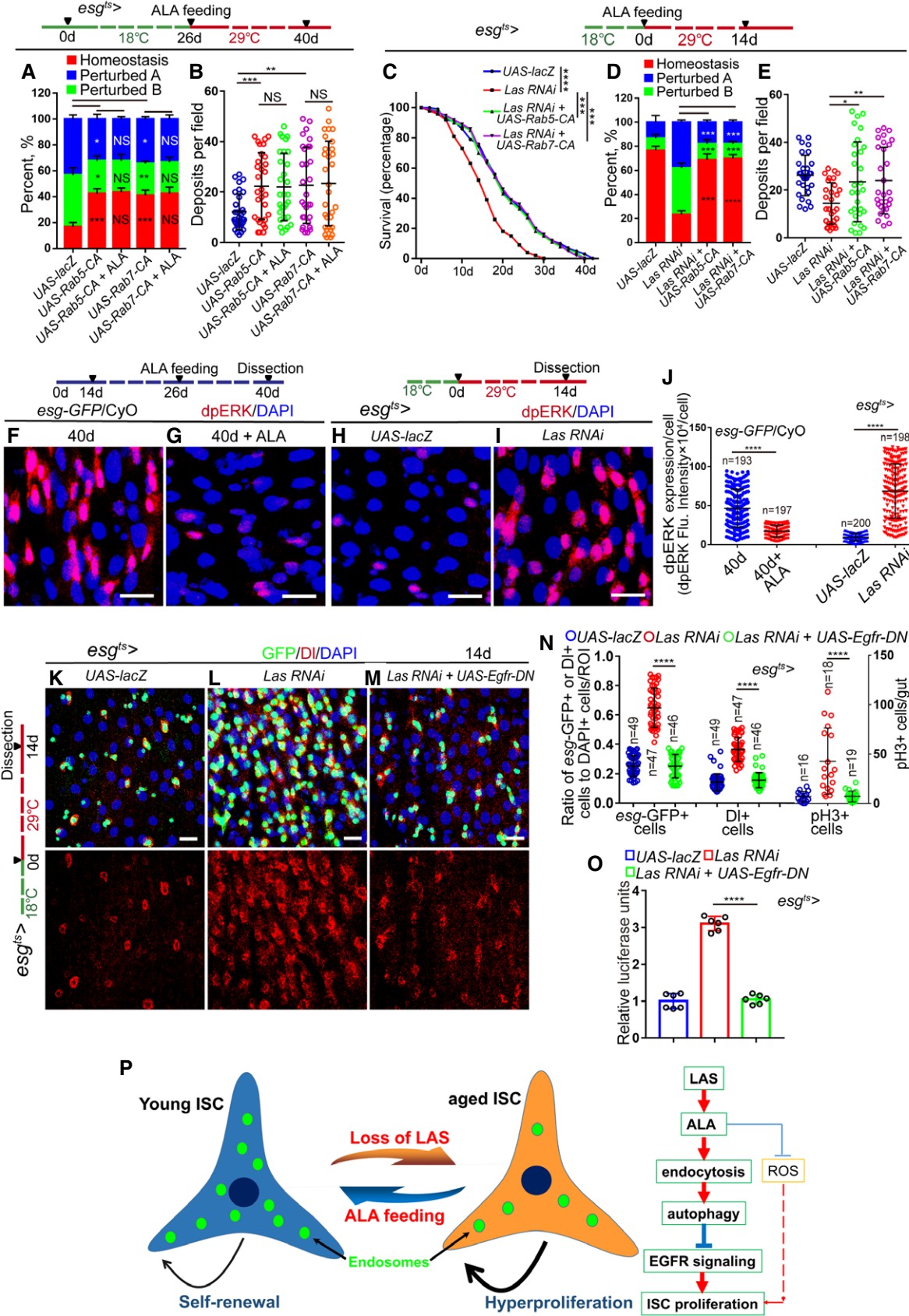

Figure 10.

◀

**Figure 10.  ALA restrains ISC overproliferation in old *Drosophila* by an endocytosis-autophagy-mediated EGFR activity counteraction.**

A       Quantification of three categories of midguts treated with the pH indicator Bromophenol blue in *Drosophila* of indicated genotypes. The numbers of quantified guts from left to right are 90, 90, 90, 90, and 90. Error bars show the SD of three independent experiments.

B       Quantification of excretion numbers of *Drosophila* with indicated genotypes. Excretions are quantified in 30 fields for each group of 12 *Drosophila*. Tests were repeated as three independent experiments.

C       Survival (percentage) of female *Drosophila* of indicated genotypes. The numbers of quantified *Drosophila*: 100 (*UAS-lacZ*), 100 (*Las RNAi*), 100 (*Las RNAi + UAS-Rab5-CA*) and 100 (*Las RNAi + UAS-Rab7-CA*). Three independent experiments were conducted

D       Quantification of three categories of midguts treated with the pH indicator Bromophenol blue of *Drosophila* with indicated genotypes. The numbers of quantified guts from left to right are 90, 90, 90, and 90. Error bars show the SD of three independent experiments.

E       Quantification of excretion numbers of *Drosophila* of indicated genotypes. Excretions are quantified in 30 fields for each group of 12 *Drosophila*. Tests were repeated as three independent experiments.

F–I    Immunofluorescence images of dpERK staining of midgut sections of the R4 region of 40-day *Drosophila* (F), 40-day *Drosophila* with ALA administration (G), 14-day *Drosophila* carrying *esg*^ts^-GAL4-driven *UAS-lacZ* (H), and *Drosophila* carrying *esg*^ts^-GAL4-driven *Las RNAi* (I). dpERK (red).

J       Quantification of the fluorescence intensity of dpERK in experiments (F–I). Each dot corresponds to one cell. *n* is indicated. The numbers of quantified guts from left to right are 11, 12, 12, and 13.

K–M   Representative images of midgut sections from the R4 region of *Drosophila* carrying *esg*^ts^-GAL4-driven expression of *lacZ* cDNA (K, control), *Las RNAi* (L), or *Las RNAi* and dominant-negative form of *EGFR* (*EGFR-DN*, M). GFP (green) and Dl staining (red) was used to visualize ISCs.

N       Quantification of *esg*-GFP^+^ cells, Dl^+^ cells, and pH3^+^ cells in experiments (K–M). *n* is indicated. The numbers of quantified guts from left to right are 16, 18, 19, 16, 18, 19, 16, 18, and 19.

O       Quantification of luciferase activity of midguts of *Drosophila* carrying *esg*^ts^-GAL4-driven expression of *lacZ* cDNA, *Las RNAi*, or *Las RNAi* and *EGFR-DN*. Error bars show the SD of six independent experiments.

P       Schematic model of the mechanism. In young flies highly expressed LAS synthesizes sufficient ALA in ISCs. ALA promotes endosome formation and maturation in ISCs, which in turn maintains the proper activation of autophagy and prevents the over activation of EGFR signaling in ISCs. As a secondary effect, ALA also prevents ISC overproliferation by scavenging excessive ROS. When flies get old, the expression of LAS dramatically reduces, which causes ALA deficiency and endosome reduction in ISCs. Ultimately, this leads to the hyperproliferation of ISCs with age.

Data information: DAPI-stained nuclei are shown in blue. Scale bars represent 10 μm (F–I, and K–M). Error bars represent SDs. *P*-values for lifespan curves (C) were calculated by the log-rank test. The statistical tests used in other panels were Student's *t*-tests. *$P < 0.05$, **$P < 0.01$, ***$P < 0.001$, ****$P < 0.0001$, and non-significant (NS) represents $P > 0.05$. See also Fig EV5.

Source data are available online for this figure.

overproliferation (Fig 10F–O). Further studies should investigate how ALA modulates the endocytosis activity during aging.

Alpha-lipoic acid is naturally located in mitochondria, where it was *de novo* synthesized by LAS from octanoic acid. The production of endogenous ALA is directly controlled by the expression level and activity of LAS. Recent studies indicated the reduction of ALA or the deficiency of LAS expression to be linked to several diseases, such as diabetes, atherosclerosis, and neonatal-onset epilepsy (Padmalayam *et al*, 2009; Krishnamoorthy *et al*, 2017). Increased ALA or LAS expression had many beneficial metabolic effects in animal models of obesity and type 2 diabetes, such as a decreasing body weight gain, and improving the plasma glucose and lipid profile (Shay *et al*, 2009; Park *et al*, 2014; Krishnamoorthy *et al*, 2017; Salehi *et al*, 2019). These results strongly suggest the existence of a strong inverse correlation between LAS downregulation and disease status. This study showed that the expression of LAS in *Drosophila* ISCs dramatically decreased upon aging. However, the mechanism of how LAS expression is downregulated in aged ISCs remains unknown. Further studies should explore how the extracellular clues incorporated intercellular signals to regulate LAS expression during aging.

# Materials and Methods

### *Drosophila* breeding and maintenance

All flies were maintained on the standard cornmeal and yeast fly food (the recipe for 1 l food is: cornmeal 50 g, yeast 18.75 g, sucrose 80 g, glucose 20 g, agar 5 g, and propionic acid 30 ml), if not specifically mentioned. Unless noted otherwise, flies were cultured at 25°C in a normal light/dark cycle. To repress the GAL4 system, the crosses were maintained at 18°C when driving temperature-sensitive GAL4-mediated RNAi or gene overexpression. When flies eclosion or at a certain age after flies eclosion, the adults were shifted to 29°C to turn on the GAL4 system, which induces RNAi or gene overexpression. Eclosed flies were incubated at 29°C for indicated days followed by dissection of midguts for immunostaining. If not specifically mentioned, we used the mated females for experiments on the *Drosophila* midguts.

### *Drosophila* lines in this study

The *w*^1118^ (BDSC3605) and *Canton-S* (BDSC64349) allele was used as the wild-type control. We did not see any difference between *esg-GFP/CyO* flies and *w*^1118^ flies, so some time we also used *esg-GFP/CyO* flies as control. The UAS-luciferase line was obtained by using stock *P{nSyb-MKII::GAL4DBDo}attP24, P{QUAS-p65AD::CaM}2/CyO; P{UAS-LUC.D}3* purchased from BDSC (61687). The use of other *Drosophila* lines is noted within the figures, legends, and text. *Drosophila* lines used in this study are listed in Table EV1. Full fly genotypes as they appear in each Figure panel are listed in Table EV2.

### Clonal analyses

To generate *Las* RNAi MARCM clones, we used *Las* RNAi and FLP/FRT-mediated mitotic recombination technique. Cross *Las RNAi* to *FRT40A* to produce flies: *FRT40A/cyo; Las RNAi/TM6B*, which was then crossed to *yw hsFLP, tub-Gal4, UAS-nls GFP/FM7; tubG80 FRT40A/CyO* to obtain *yw hsFLP, tub-Gal4, UAS-nls GFP/ + ; tubG80 FRT40A/ + ; Las RNAi/+* flies. Crosses were maintained at 25°C. To

induce clones, 2- to 3-day-old adult flies after eclosion were subjected to 1 h heat shock in the 37°C water bath. After heat shock, flies were kept at 25°C, and then these flies were dissected and observed at 10 days after clone induce (ACI).

## Midgut dissection for RNA-seq

The adult midguts (R1–R5) of flies were dissected from the whole guts by removing foreguts, hindguts, Malpighian tubules, and trachea. Total RNA was prepared from more than 50 female midguts (R1–R5) for RNA-seq.

Dissected midguts were immediately frozen on dry ice and used to prepare the total RNA using isothiocyanate-alcohol phenyl-chloroform. The whole previous sequencing was carried out by Berry Genomics Corporation (China). Sequence platform is based on novaseq 6000 (Illumina, San Diego, US) using a 150 bp paired-end run resulting in over 20 million reads per sample.

Raw RNA-seq data were processed initial quality control, including filtering out low-quality reads and cutting off adapter sequences by Trimmomatic (v 0.39). Clean fastq files comprised paired-end reads with read length of 150 bp, and were secondarily quality controlled by FastQC (v0.11.8, http://www.bioinformatics.babraham.ac.uk/projects/fastqc/). Reads were aligned to the *Drosophila* reference genome downloaded from Ensembl BDGP6 release-98 (ftp://ftp.ensembl.org/pub/release-98/) using HISAT2 (Kim *et al*, 2015) (v 2.1.0). Aligned reads sam files were then assembled into bam files sorted by chromosome position by SAMtools (v 1.9). Gene raw counts matrix per sample was calculated by StringTie (v 1.3.5). Differentially expressed genes were determined using R package DESeq2 (Love *et al*, 2014) (v 1.24.0) with default parameters. Differentially expressed genes were deemed if *P*-value < 0.05 following a Benjamini and Hochberg correction for multiple hypothesis testing (default parameter of DESeq2). Downstream pathway analysis was then performed via R package clusterProfiler (Yu *et al*, 2012) (v 3.8.1). Downstream analysis software is supported by R (v 3.5.3). (https://www.r-project.org/).

## FACS and qRT–PCR

One hundred female midguts were dissected into ice-cold DEPC-PBS and incubated with 1 mg/ml Elastase (Sigma, cat. no. E0258) diluted with DEPC-PBS for 1 h at 25°C, during which the sample was softly mixed every 15 min by pipetting and inverting five times. Dissociated midguts were pelleted at 400 *g* for 20 min at 4°C, re-suspended in 0.5 ml ice-cold DEPC-PBS, filtered with 70 μm filters (Biologix), and sorted using a FACS Aria II sorter (BD Biosciences). Using w1118 midgut to set the fluorescence gate, GFP$^+$ cells in the midgut of *esg*-GFP flies were sorted out. For each of the three biological replicates, about 40,000 *esg*-GFP$^+$ cells were sorted. Then, the total RNA was harvested using the Arcturus PicoPure RNA Isolation Kit (Applied Biosystems) based on the manufacturer's protocol. We synthesized cDNA using the PrimeScript RT reagent Kit (TaKaRa). All total RNA was used for reverse transcription with oligo dT. Then, the first-strand cDNA was diluted 50 times with distilled water and further used in real-time PCR. Real-time PCR was performed in at least triplicate for each sample using SYBR Green (genestar) on a QuantStudio 5 System (Thermo Fisher Scientific). Expression values of RT–qPCR were calculated using the $2^{-\Delta\Delta C_T}$

method, and the relative expression was normalized to Rp49. The expression of control samples was further normalized to 1. Primer sequences used for qPCR are available upon reasonable request.

## Guts lysate preparation and Western blotting analyses

The dissected midguts of flies were frozen with liquid nitrogen in RIPA Lysis Buffer (P0013B, Beyotime Biotechnology) with protease inhibitors. The midguts were homogenized with grinding rod and then were placed on ice for 30 min. By centrifugation at 13,500 *g* for 15 min, the lysate supernatant was collected as total protein to measure the concentrations of proteins using the BCA kit (Pierce™ Rapid Gold BCA Protein Assay Kit, Thermo, A53226). Later, total protein from each sample mixed with 5× loading buffer was boiled for 10 min and analyzed by 10–12% SDS–PAGE. After that, the gels were transferred onto polyvinylidene difluoride (PVDF) membranes. After blocking with 5% non-fat milk in TBST for 1 h and washing three times (5 min each), the PVDF membranes were incubated with primary antibodies overnight at 4°C and washed four–five times with TBST at room temperature (5 min each). The PVDF membranes were shaken and incubated at room temperature for 1 h with the horseradish peroxidase-labeled secondary antibody. The antibodies used for Western blotting were listed in the Table EV3. The used secondary antibodies were horseradish peroxidase-conjugated goat anti-mouse (Beyotime Biotechnology, #A0216, 1:1,000) or goat anti-rabbit (Thermo, #A0208, 1:1,000).

## Immunofluorescence microscopy for *Drosophila* tissue

Adult *Drosophila* midguts were dissected in cold PBS and were fixed in 4% EM-grade paraformaldehyde fixation buffer (100 mM glutamic acid, 25 mM KCl, 20 mM MgSO$_4$, 4 mM Na$_2$HPO$_4$, 1 mM MgCl$_2$, pH 7.4) for 30–60 min followed by washing in the wash buffer (PBS plus 0.3% Triton X-100) for three–five times, 10 min each. Samples were then blocked in the wash buffer with 0.5% BSA for 30 min followed by incubation with primary antibodies at 4°C overnight. All primary antibodies were pre-absorbed by 4% paraformaldehyde-fixed embryos. After washing three times (10 min each), the midguts were incubated with secondary antibodies and DAPI for 2 h at room temperature followed by the same washing steps above.

The sources and dilutions of all primary antibodies used in this study are listed in Table EV3. The secondary antibodies (Alexa 488, Alexa 568, and Alexa 647 from MolecularProbes/Invitrogen) were diluted and used at 1:2,000. The final concentration of 4,6-diamidino-2-phenylindole (DAPI; Sigma) is 1 μg/ml.

All immunofluorescence images were acquired by Leica TCS-SP8 confocal microscope. Images for each set of experiments were acquired as confocal stacks using the same setting. The images were assembled using Adobe Photoshop and Adobe Illustrator CS3. The number of midgut cells in all quantifications was counted using a Leica DM6-B microscope.

## Lysotracker staining

Midguts were dissected and incubated with 0.8 μM Lysotracker (Invitrogen, L7528) for 5 min and Hoechst for 10 min, washed three times for 2 min each with PBS, and imaged immediately by confocal

microscopy. All washes and incubations were conducted at room temperature and in the dark.

## Luciferase assays

Luciferase was measured using the Firefly Luciferase Reporter Gene Assay Kit (Beyotime Biotechnology, Jiangsu, China, RG051S). Fifteen female adult midguts were dissected in cold PBS and frozen immediately on liquid nitrogen. And then, samples were homogenized in 50 μl Luciferase Reporter Gene Assay Lysis Buffer provided by the manufacturer. Sample extracts were subsequently centrifuged at 13,000 *g* for 10 min at 4°C to obtain samples for testing. Samples were collected over a series of days and stored at −80°C until six independent samples were collected for each genotype. Luciferase activity was measured based on the protocols provided by the manufacturer.

## Bromophenol blue treatment

Bromophenol blue assay was performed as previously described (Li *et al*, 2016). One hundred microliter of 2% Bromophenol blue sodium (pH indicator, Sigma, B5525) was added to a food vial and then poke 4–6 holes in the food using pipet tip to allow full absorption. Twelve hours later, images were taken immediately after each gut was dissected.

## Cafe assay and fly excretion measurement

The Cafe and fly excretion assays were performed as previously described (Cognigni *et al*, 2011; Deshpande *et al*, 2014; Li *et al*, 2016).

## Alpha-lipoic acid, *N*-acetylcysteine amide, and spermidine treatment

Alpha-lipoic acid (aladdin, D118666), *N*-acetylcysteine amide (AD4, aladdin, N170064), and spermidine (aladdin, S107071) were dissolved in DMSO, and then added to regular food medium. Flies were collected and distributed equally into food vials with ALA, AD4, or spermidine medium.

## *Drosophila* survival tests

For survival tests, 100 female flies (1–2 days old) of the same genetic background were collected and distributed equally into four vials with regular food medium or with ALA medium. At the same time, 10 males (1–2 days old) were added to each vial to ensure that the females were mated. Female flies that were still alive were counted every 2 day. The viability tests were repeated as three independent experiments.

## Liquid chromatography-electrospray ionization-mass spectrometry analysis

### Chemicals and reagents:

Alpha-lipoic acid (D118666), 8-aminooctanoic acid (an internal standard for ALA, A165345) were obtained from Aladdin (shanghai, china). The methanol and acetonitrile (ACN) of LC-ESI-MS/MS

grade were purchased from Sigma-Aldrich (St. Louis, USA). Chloroform, glacial acetic acid, and ammonia of AR grade were obtained from Guangzhou Chemical Reagent Factory (Guangzhou, China).

### Extraction of ALA from *Drosophila* midguts

The adult midguts (R1–R5) of flies were dissected from the whole guts by removing foreguts, hindguts, Malpighian tubules, and trachea. To get these midguts, 300 female adult *Drosophila* were collected at a fixed time in the day (2:00 pm to 4:00 pm). Their midguts were dissected in 1.5 ml centrifuge tube with cold PBS and centrifuged at 13,000 *g* for 10 min at 4°C. Remove PBS as cleanly as possible and accurately determine the weight of the dissected midgut (Weigh the empty centrifuge tube in advance). And then, midguts were crushed by using a tissue grinder in 200 μl RIPA Lysis Buffer (P0013B, Beyotime Biotechnology) and incubated for 30 min on ice. Chloroform (500 μl) was added to the extracts, thoroughly shaken for 30 s, and centrifuged at 13,000 *g* for 10 min at 4°C. The lower organic (chloroform) phase was removed in another centrifuge tube. The upper water phase and intermediate solid phase were extracted with chloroform for three times. The organic (chloroform) phases were combined and dried at the room temperature. Ninety microliter acetonitrile and 10 μl 8-aminooctanoic acid (internal standard, 10 μg/ml dissolved in water) was added, and the cleared supernatant (10 μl) was injected into LC-ESI-MS/MS mass spectrometer (TripleTOF™ 5600+, AB SCIEX, USA) under negative ESI multiple reaction monitoring (MRM) to determine ALA concentration.

### Calibration curves

The standard solution was prepared at the concentration of 0.5, 1, 2, 5, 10, 20, 100, 200, and 1,000 ng/ml, respectively, and 8-aminooctanoic acid (internal standard) was added to the final concentration of 1 μg/ml. Ten microliter solution was injected into LC-ESI-MS/MS mass spectrometer (TripleTOF™ 5600+, AB SCIEX, USA). A calibration curve was prepared from the relative area ratio of the standard and the internal standard (*y*-axis) to the concentration of standard (*x*-axis). The calibration curve had good linearity ($r = 0.9998$) in the range of 0.5–1,000 ng/ml. The linear regression equation is as follows: $y = 0.0298x + 0.0341$; $r = 0.9998$.

### Determination of ALA

Ten microliter extracted sample was injected into LC-ESI-MS/MS mass spectrometer (TripleTOF™ 5600+, AB SCIEX, USA) under negative ESI MRM to obtain the relative area ratio of the ALA and the internal standard and calculate the amount of ALA in the sample. Liquid chromatography was performed by HPLC (SHIMADZU, LC20A) with Poroshell 120 LC column (Bonus-BP 3.0 × 100 mm, 2.7 μm; Agilent). Mobile phase A, methanol: 0.1% glacial acetic acid (add ammonia to pH 4.5) = 55: 35. Mobile phase B, 100% acetonitrile. Mobile phase A: mobile phase B = 90: 10. Flow rate, 0.25 ml/min with 5 mM ammonium formate for and 100% methanol for mobile phase B. Mass spectrometry was performed by a LC-ESI-MS/MS mass spectrometer (TripleTOF™ 5600+, AB SCIEX, USA) under negative ESI MRM using parameters for ALA ($m/z$ 205.0->171.0). Mass spectrometer were as follows: ion source gas1 55 psi; ion source gas2 55 psi; curtain gas 35 psi; temperature 500°C; collision energy 40 V; collision energy spread 20 V and declustering potential 80 V, ion spray voltage floating 5,500 V.

## Quantification and statistical analysis

Data are presented as the mean standard deviation (SD) from at least three independent experiments unless otherwise specified in the method or figure legends. Statistical significance was determined using the two-tailed Student's *t*-test unless otherwise specified in the figure legends. Significance is timely stated in text or figure legends. For all tests, a $P < 0.05$ was considered statistically significant.

## Fluorescence intensity statistics

Immunofluorescence imaging results were analyzed based on z stacks acquired with Leica TCS-SP8 confocal microscope. Fluorescence intensity of the region of interest (ROI) was calculated by ImageJ software. The methods involved are as below:

Open image: File → Open. Split Channels: image→ color →Split Channels. Keep the channel to be calculated and delete other channels. Set scale to make sure the unit of length is Pixel: Analyze → Set scale →click to remove scale. Set measurements. Analyze →Set Measurements. Tick the boxes marked "Area," "Integrated Density," and "limit to threshold." Then click on the "OK" button. Select the ROI using any of the drawing/selection tools and select a smaller region around the ROI as background. Select different ROI and background regions by using ROI Manager. Tick the boxes marked "measure" to calculate the integrated density. Integrated Density = Integrated Density of ROI − Integrated Density of background region/Area of background region × Area of ROI.

## Band Gray Value statistics of Western blotting

Band Gray Value statistics of Western blotting analyses were calculated by ImageJ software. The methods involved are as below:

Open image: File →Open. Invert: Edit →invert. Set scale to make sure the unit of length is Pixel: Analyze →Set scale →click to remove scale. Set measurements. Analyze →Set Measurements. Tick the boxes marked "Area," "Mean gray value," and limit to the threshold. Then click on the "OK" button. Select the region of the band using any of the drawing/selection tools and select a smaller region around the band as background. Select the different regions of band and background region by using ROI Manager. Tick the boxes marked "measure" to calculate the mean gray value. Band Gray Value = mean gray value of band region × Area of band region − mean gray value of background region × Area of the background region.

# Data availability

The RNA-seq data that support the findings of this study have been deposited in the Sequence Read Archive (SRA) under BioProject ID PRJNA579055 (https://www.ncbi.nlm.nih.gov/sra/?term = PRJNA579 055). Source data for Fig 5A–C have been provided as Datasets EV1 and EV2. The R (version 3.5.3, download from https://www.r-project.org/) was used for downstream analysis of RNA-seq. The custom ImageJ software that used for immunofluorescent staining and Western blotting quantification is available at https://imagej.nih.gov/ij/.

**Expanded View** for this article is available online.

## Acknowledgements

We thank BDSC, VDRC, DGRC, and Tsinghua Fly Center for fly strains, DSHB for antibodies. This work was supported by the National Key Research and Development Program Stem Cell and Translational Research Key Projects (2018YFA0108301), the National Natural Science Foundation of China (31622031, 31671254, and 91749110) (H.C.), the Guangdong Natural Science Funds for Distinguished Young Scholar (2016A030306037) (H.C.), and the National Clinical Research Center for Geriatrics, West China Hospital, Sichuan University.

## Author contributions

Conceptualization: HC and GD; Methodology: GD, YQ, ZZ, JZ, HC; Investigation: GD, YQ, ZZ, YL, JZ, XL, ZL; Writing-original draft: GD, YQ, YL, HC; Writing-review & editing: GD, HC; Funding acquisition: HC; Resources: HC; Supervision: HC.

## Conflict of interest

The authors declare that they have no conflict of interest.

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
