## [Review Process File · EMBO Reports]

Lipoic acid rejuvenates aged intestinal stem cells by preventing age-associated endosome reduction

Haiyang Chen, Gang Du, Yicheng Qiao, Zhangpeng Zhuo, Jiaqi Zhou, Xiaorong Li, Zhiming Liu, and Yang Li

DOI: [10.15252/embr.201949583](https://doi.org/10.15252/embr.201949583)

Corresponding author(s): Haiyang Chen (chenhy87@mail.sysu.edu.cn)

Review Timeline:

Submission Date:	2nd Nov 19
Editorial Decision:	10th Dec 19
Revision Received:	8th Mar 20
Editorial Decision:	11th May 20
Revision Received:	15th May 20
Accepted:	29th May 20

Editor: Deniz Senyilmaz Tiebe

Transaction Report:

Dear Prof. Chen,

Thank you for submitting your manuscript for consideration by EMBO Reports. Three referees agreed to review your manuscript. So far, we have received two referee reports that are copied below. Given that both referees are in fair agreement that you should be given a chance to revise the manuscript, I would like to ask you to begin revising your study along the lines suggested by the referees.

Please note that this is a preliminary decision made in the interest of time, and that it is subject to change should the third referee offer very strong and convincing reasons for this. As soon as we receive the final report on your manuscript, we will forward it to you as well.

As you can see, the referees express interest in the proposed effects of lipoic acid on intestinal stem cell aging. However, they also raise a number of concerns that need to be addressed to consider publication here. I find the reports informed and constructive, and believe that addressing the concerns raised will significantly strengthen the manuscript.

Given these constructive comments, we would like to invite you to revise your manuscript with the understanding that the referee concerns (as in their reports) must be fully addressed and their suggestions taken on board. Please address all referee concerns in a complete point-by-point response. Acceptance of the manuscript will depend on a positive outcome of a second round of review. It is EMBO reports policy to allow a single round of revision only and acceptance or rejection of the manuscript will therefore depend on the completeness of your responses included in the next, final version of the manuscript.

1. A data availability section providing access to data deposited in public databases is missing (where applicable).
2. Your manuscript contains statistics and error bars based on $n=2$ or on technical replicates. Please use scatter plots in these cases.

Supplementary/additional data: The Expanded View format, which will be displayed in the main HTML of the paper in a collapsible format, has replaced the Supplementary information. You can submit up to 5 images as Expanded View. Please follow the nomenclature Figure EV1, Figure EV2 etc. The figure legend for these should be included in the main manuscript document file in a section called Expanded View Figure Legends after the main Figure Legends section. Additional Supplementary material should be supplied as a single pdf labeled Appendix. The Appendix includes a table of content on the first page with page numbers, all figures and their legends. Please follow the nomenclature Appendix Figure Sx throughout the text and also label the figures according to this nomenclature. For more details please refer to our guide to authors.

2) individual production quality figure files as .eps, .tif, .jpg (one file per figure).

3) a .docx formatted letter INCLUDING the reviewers' reports and your detailed point-by-point responses to their comments. As part of the EMBO Press transparent editorial process, the point-by-point response is part of the Review Process File (RPF), which will be published alongside your paper. For more details on our Transparent Editorial Process, please visit our website: <https://www.embopress.org/page/journal/14693178/authorguide#transparentprocess>
You are able to opt out of this by letting the editorial office know (emboreports@embo.org). If you do opt out, the Review Process File link will point to the following statement: "No Review Process File is available with this article, as the authors have chosen not to make the review process public in this case."

4) a complete author checklist, which you can download from our author guidelines (<<http://embor.embopress.org/authorguide>>). Please insert information in the checklist that is also reflected in the manuscript. The completed author checklist will also be part of the RPF.

5) Please note that all corresponding authors are required to supply an ORCID ID for their name upon submission of a revised manuscript (<<https://orcid.org/>>). Please find instructions on how to link your ORCID ID to your account in our manuscript tracking system in our Author guidelines (<<http://embor.embopress.org/authorguide>>).

6) We replaced Supplementary Information with Expanded View (EV) Figures and Tables that are collapsible/expandable online. A maximum of 5 EV Figures can be typeset. EV Figures should be cited as 'Figure EV1, Figure EV2' etc... in the text and their respective legends should be included in the main text after the legends of regular figures.

- For the figures that you do NOT wish to display as Expanded View figures, they should be bundled together with their legends in a single PDF file called *Appendix*, which should start with a short Table of Content. Appendix figures should be referred to in the main text as: "Appendix Figure S1, Appendix Figure S2" etc. See detailed instructions regarding expanded view here: <<http://embor.embopress.org/authorguide#expandedview>>.

7) We would also encourage you to include the source data for figure panels that show essential data.

Numerical data should be provided as individual .xls or .csv files (including a tab describing the data). For blots or microscopy, uncropped images should be submitted (using a zip archive if multiple images need to be supplied for one panel). Additional information on source data and instruction on

how to label the files are available <<http://embor.embopress.org/authorguide#sourcedata>>.

8) Our journal encourages inclusion of *data citations in the reference list* to directly cite datasets that were re-used and obtained from public databases. Data citations in the article text are distinct from normal bibliographical citations and should directly link to the database records from which the data can be accessed. In the main text, data citations are formatted as follows: "Data ref: Smith et al, 2001" or "Data ref: NCBI Sequence Read Archive PRJNA342805, 2017". In the Reference list, data citations must be labeled with "[DATASET]". A data reference must provide the database name, accession number/identifiers and a resolvable link to the landing page from which the data can be accessed at the end of the reference. Further instructions are available at <<http://embor.embopress.org/authorguide#datacitation>>.

9) Before submitting your revision, primary datasets (and computer code, where appropriate) produced in this study need to be deposited in an appropriate public database (see <<http://embor.embopress.org/authorguide#dataavailability>>).

The accession numbers and database should be listed in a formal "Data Availability " section (placed after Materials & Method) that follows the model below. Please note that the Data Availability Section is restricted to new primary data that are part of this study.

Data availability

10) Regarding data quantification, please ensure to specify the name of the statistical test used to generate error bars and P values, the number (n) of independent experiments underlying each data point (not replicate measures of one sample), and the test used to calculate p-values in each figure legend. Discussion of statistical methodology can be reported in the materials and methods section, but figure legends should contain a basic description of n, P and the test applied. Please note that error bars and statistical comparisons may only be applied to data obtained from at least three independent biological replicates. Please also include scale bars in all microscopy images.

I look forward to seeing a revised version of your manuscript when it is ready. Please let me know if you have questions or comments regarding the revision.

Yours sincerely,

Deniz Senyilmaz Tiebe

Deniz Senyilmaz Tiebe, PhD
Editor
EMBO Reports

Referee #1:

In this paper, Du et al. report that feeding *Drosophila* alpha-lipoic acid (ALA) prevents age-associated intestinal stem cell (ISC) hyperproliferation. Further, they present evidence that reducing ALA synthesis in ISCs can promote midgut dysplasia that is similar to that which occurs with aging. Genetic tests also suggest that ALA synthesis may decline in ISCs during aging. These effects are attributed to the ability of ALA to sustain high levels of autophagy and endocytosis, and compelling data is provided to support this conclusion. All of these discoveries are quite novel and exciting, and will be appreciated in the intestinal stem cell field. The paper is well organized and clearly written, and contains a large amount of very good quality genetics data. The effects that are shown are surprisingly large, robust, and consistent (but see below). The paper's main weaknesses in my opinion are that: 1) ALA levels are not measured directly in ISCs or the gut during aging, or after the various genetic manipulations; and 2) nothing is known about how ALA would affect endocytosis, autophagy, or other metabolic processes that might impact ISC functional decline. I believe the former point should be addressed here if possible; the later issue is a much larger one, endemic to the field, and can await further investigation. In addition, we have a number of specific issues the authors should try to address as they revise this interesting manuscript:

Specific points follow below:

1. Does ALA feeding (or LAS knockdown) affect ISC mitosis or promote differentiation? In either case, age-dependent dysplasia would be affected. A MARCM clone assays (LASRNAi or LAS^{-/-}) could be done to distinguish these possibilities.
2. Is LAS overexpression sufficient to raise ALA levels, sustain autophagy, suppress dysplasia, and extend lifespan. This obvious test is missing from the study and should be added if possible.
3. The genetics data suggesting that ALA promotes autophagy and endocytosis are compelling. However, the data on autophagosome formation itself are somewhat weak and need to be improved. We suggest that the authors: a) provide more example pictures for Fig 5a-e, in the supplement; b) include the lysotracker data (Fig EV3) in the main Fig 5; c) provide quantifications and color separations for the lysotracker data; c) explain (if possible) why lysotracker was affected in all cells when las-RNAi was only expressed in esg⁺ cells (Fig EV3d-e).
4. Some minor grammatical issues need to be fixed.
5. Most of the graphical data presented looks suspiciously robust and consistent. This is worrisome

as most studies with the *Drosophila* midgut show considerable animal-to-animal variation in most cellular responses, especially during aging. Hence, I believe the authors need to present strong evidence that their data were collected fairly using unbiased methods. This could include statements about which data were collected in a blinded fashion, and submission of raw datasets. The bar graphs (e.g. Fig 7B) might better be presented as dot-plots (e.g. Fig 7J) to illustrate actual variation. Examples of the "too good to be true" data I find worrisome can be found in most of the graphs: e.g. Fig 3N, 5KLQRTW, 6RSYZAA, 7BE. The surprisingly consistent quality of this data raises ethical concerns. Hopefully this concern is unfounded and the data quality simply reflects the excellent skills of these researchers, but proof must be given.

6. For all the lifespan data (Fig 2G, 5U, 7C), please confirm that mated females were used, as in the other experiments. Also, p values should be presented. Please present the other 2 replicates for each lifespan test in the supplement.

7. In Fig 6, the authors use Rab5-CA or Rab7-CA to upregulated endocytosis. They need to present data and/or citations showing that these overexpressions really do upregulate endocytosis.

Referee #3:

Du et al. present interesting findings related to lipoic acid, intestinal homeostasis and aging in *Drosophila*. There are many strengths to this manuscript, including the large amount of genetic experiments that provide mechanistic insight. The most exciting aspect of the study is that feeding ALA to aged flies can rejuvenate aspects of ISC aging. This is very interesting. On the whole, the experiments are carefully carried out, however, I do have some technical concerns and suggestions to improve the manuscript.

- 1) It is clear that ISC homeostasis can be improved by ALA feeding in 26day old flies. It would increase the impact of the paper if it were shown that day 26 feeding could prolong lifespan. In addition, multiple 'lab strains' should be tested. w1118 is a notoriously unhealthy line
- 2) It is critical to assay feeding behavior in control flies fed ALA. This is missing. If ALA impaired feeding in control 26d old flies that would be a confounding factor
- 3) It is great that the authors sought to examine autophagic activity in aged ISCs. However, my understanding is that the reagent that was used ATg8 tandem repeat is under UAS-control. Hence, in using this reagent aren't the authors overexpressing Atg8? If so, this is a confounding factor, no?

Dear Dr. Tiebe,

Many thanks for handling our manuscript. All reviewers have commented on the originality and importance of our findings. We have address reviewers' comments by both editing the text of the manuscript and performing additional experiments. Below please find our point-by-point response (*italic maroon color*) to each of the comment (copied in full in black). We have also provided references cited at the end of the response.

We hope that our revision is satisfactory.

Best wishes!

Haiyang Chen

(On behalf of all authors)

Referee #1:

In this paper, Du et al. report that feeding *Drosophila* alpha-lipoic acid (ALA) prevents age-associated intestinal stem cell (ISC) hyperproliferation. Further, they present evidence that reducing ALA synthesis in ISCs can promote midgut dysplasia that is similar to that which occurs with aging. Genetic tests also suggest that ALA synthesis may decline in ISCs during aging. These effects are attributed to the ability of ALA to sustain high levels of autophagy and endocytosis, and compelling data is provided to support this conclusion. All of these discoveries are quite novel and exciting, and will be appreciated in the intestinal stem cell field. The paper is well organized and clearly written, and contains a large amount of very good quality genetics data. The effects that are shown are surprisingly large, robust, and consistent (but see below). The paper's main weaknesses in my opinion are that: 1) ALA levels are not measured directly in ISCs or the gut during aging, or after the various genetic manipulations; and 2) nothing is known about how ALA would affect endocytosis, autophagy, or other metabolic processes that might impact ISC functional decline. I believe the former point should be addressed here if

possible; the later issue is a much larger one, endemic to the field, and can await further investigation. In addition, we have a number of specific issues the authors should try to address as they revise this interesting manuscript:

Response:

We thank this reviewer for appreciating the novelty and importance of our findings. We have performed LC-ESI-MS/MS analyses and measured the ALA levels in the midguts of flies during aging (See Figs 1L-O, and Figs EV2A-E) and in the midguts of flies with Las depleted (carrying Act5C^{ts}-GAL4>Las RNAi; See Figs 2A-C). These data indicate that ALA indeed downregulates in midguts of flies upon aging and of flies with LAS knockdown. We agree with this reviewer that we didn't provide the mechanism of how ALA affect the endocytosis and autophagy processes in ISCs. We thank that this reviewer allows us not to address this issue in this manuscript but allow us to investigate it in the future. We will definitely follow this question to do further investigation in the future.

We appreciate the critique from this reviewer and have both edit the text and added additional data as described in detail below.

Specific points follow below:

1. Does ALA feeding (or LAS knockdown) affect ISC mitosis or promote differentiation? In either case, age-dependent dysplasia would be affected. A MARCM clone assays (LAS RNAi or LAS^{-/-}) could be done to distinguish these possibilities.

Response:

We have analyzed Las-RNAi MARCM clones. We found that ISCs with Las RNAi can forms normal MARCM clones, which indicated that these Las-depleted ISCs can initiate mitosis (See Figs 2I-J). Based on Pdm1(which labels differentiated enterocytes (ECs)) staining, deletion of LAS did not show obvious ISC differentiation defects (See Figs 2I-K). However, we indeed found that the average size of Las-RNAi MARCM clones was obviously bigger compared to that of the control clones (See Figs 2I-J and 2L). These suggested that LAS does not regulate ISC differentiation but regulate the mitotic rate of ISC proliferation. In the revised manuscript, we have included these new data.

2. Is LAS overexpression sufficient to raise ALA levels, sustain autophagy, suppress dysplasia, and extend lifespan. This obvious test is missing from the study and should be added if possible.

Response:

We have overexpressed LAS using Act5C-GAL4^{ts} (an ubiquitously expressed GAL4). But we found that only overexpression of LAS was not sufficient to raise the ALA level in fly midguts (see in the images below). We think the possible reason maybe because the synthesis of ALA need not only LAS but also other key participants, such as the substrates and coenzymes. So depletion of LAS could lead to the disruption of LAS production, but overexpression of LAS could not induce the production of more LAS in flies. Also, simple overexpression of LAS in ISCs/EBs could not induce autophagy or suppress dysplasia (see in the images below)[Figures for referees not shown.]. Moreover, we found that overexpression of LAS did not extend lifespan but caused a shorter lifespan of flies (see in the images below). We think the possible reason maybe because overexpression of LAS will lead to cellular toxicity or induce other un-known enzymatic reactions caused lifespan shorten.

3. The genetics data suggesting that ALA promotes autophagy and endocytosis are compelling. However, the data on autophagosome formation itself are somewhat weak and need to be improved. We suggest that the

authors: a) provide more example pictures for Fig 5a-e, in the supplement; b) include the lysotracker data (Fig EV3) in the main Fig 5; c) provide quantifications and color separations for the lysotracker data; c) explain (if possible) why lysotracker was affected in all cells when las-RNAi was only expressed in esg+ cells (Fig EV3d-e).

Response:

We appreciate these comments and suggestions. a) We have provided more example pictures (for the original Figs 5A-E (Figs 6F-H and Figs 6O-P in the revised manuscript)) to show that ALA promotes autophagy process in the supplementary files (See Figs EV4A-E). b) Since this review suggested us to provide quantifications for the lysotracker data, we have re-performed these experiments using the esg-GFP reporter to restrain the boundary of ISCs/EBs (esg+ cells) and quantified the lysotracker signal restrained in the GFP+ regions. In the revised manuscript, these quantification data and images with color separated lysotracker staining were included in Figs 6A-E and Figs 6K-N. c) When we did the lysotracker staining, we indeed noticed the interesting phenomenon that lysotracker signal reduced in all midgut epithelial cells but not just in LAS-depleted esg+ cells. We think the possible reason is that depletion of LAS in esg+ cells led to the whole midgut (which has a high turnover rate and maintains by ISCs) undergo premature aging, which mimicked the old midguts and exhibited a global reduction of autophagy activity in the whole midgut epithelia. We have discussed this possibility in the revised manuscript.

4. Some minor grammatical issues need to be fixed.

Response:

Thanks for reminding. We have carefully checked our text and fixed the grammatical errors that we could find.

5. Most of the graphical data presented looks suspiciously robust and consistent. This is worrisome as most studies with the Drosophila midgut show considerable animal-to-animal variation in most cellular responses, especially during aging. Hence, I believe the authors need to present strong evidence that their data were collected fairly using unbiased methods. This could include statements about which data were collected in a blinded fashion, and submission of raw datasets. The bar graphs (e.g. Fig 7B) might better be presented as dot-plots (e.g. Fig 7J) to illustrate actual variation. Examples of the "too good to be true" data I find worrisome can be found in most of the graphs: e.g. Fig 3N, 5KLQRTW, 6RSYZAA, 7BE. The surprisingly consistent quality of this data raises ethical concerns. Hopefully this concern is unfounded and the data quality simply reflects the excellent skills of these researchers, but proof must be given.

Response:

We appreciate these comments and suggestions. We absolutely agree with this reviewer that the *Drosophila* midgut show considerable animal-to-animal variation during aging. In fact, in most of our experiment, they indeed showed considerable animal-to-animal variation. We guess that this reviewer did not see these variations in our graphical data may be because these reasons that we listed below. 1) We quantified **the ratios** of esg-GFP+ cells (or DI + cells) to total cells per ROI in midguts (such as the original Fig 5K, 5Q and 6R, 6Z, etc. (Fig EV4H, Fig7J and Figs 8R, 8Z, etc. in the revised manuscript); which tend to show smaller difference and variation between samples, such as the quantification data in references [1-5]) but **not the numbers** of esg-GFP+ cells or DI + cells per ROI in the guts (which tend to show bigger difference and variation between samples, such as the quantification data in references [6-9]). This difference between these two ways of quantification of esg-GFP+ cells (or DI + cells) is whether considers the amount change of the total cells in midguts upon aging or under stress conditions. We believe the ratio of esg-GFP+ cells (or DI + cells) to total cells per ROI in midguts is more accurate to indicate the real abundance of esg-GFP+ cells (or DI + cells) in midguts under different conditions. 2) When we quantified the luciferase activity of *Drosophila* midguts (such as the original Figs 3N, 5L, 5R, 6S, 6Y, and 6AA (Figs 4N, EV4I, 7K, 8S, 8Y, and 8AA in the revised manuscript)), we measured fly groups (15 flies as a group; see the detail in the methods) but not measured a single fly. As we know, the variation between each fly (animal-to-animal variation) is relative big, however, the variation between each fly group (a group contains 15 flies) is not that big. 3) We used the Standard Error of the Mean (SEM; a measure of precision for an estimated population mean) but not the standard deviation (SD; a measure of data variability around mean of a sample of population) to show the error bars in our graphical data. The SEM will become smaller when n (the number of quantified samples) is bigger. But, the SD will not change along with n.

As this reviewer suggested, dot-plots is a better way to illustrate actual variation. In the revised manuscript, all the Error bars of our graphical data have been replaced to represent SDs. And, all the bar graphs in our manuscript have been presented as dot-plots to illustrate actual variation. From these dot-plots, we can clearly see that the quantified samples showed reasonable animal-to-animal variation in most cases. We declare that all the quantification data that we collected were in a blinded fashion and provided more technical detail in the revised methods. We have provided all of our raw datasets for the graphical charts in our manuscript in the Expanded view flie: Source datasets. Hope that our revision and raw datasets is satisfactory.

6. For all the lifespan data (Fig 2G, 5U, 7C), please confirm that mated females were used, as in the other experiments. Also, p values should be presented.

Please present the other 2 replicates for each lifespan test in the supplement.

Response:

We confirm that mated females were used for all the lifespan data. We apologize for not describing it clearly. In the revised methods, we have described it more in detail. We have presented the other 2 replicates for each lifespan test in the Fig EV2H (relevant to Fig 3G (original Fig 2G)), Fig EV4M (relevant to Figure 7N (original Fig 5U)), and Fig EV5O (relevant to Fig 9C (original Fig 7C)) and added p values for all of our lifespan tests in the revised manuscript.

7. In Fig 6, the authors use Rab5-CA or Rab7-CA to upregulated endocytosis. They need to present data and/or citations showing that these overexpressions really do upregulate endocytosis.

Response:

Overexpress of constitutively activated Rab5 (Rab5-CA) has been demonstrated to stimulate endocytosis in various models [10, 11]. The Drosophila lines, Rab5-CA [12-14] and Rab7-CA [13, 15, 16], have been widely used in the study of endocytosis. We have added these references in the revised manuscript. Also, we found that overexpression of either Rab5-CA or Rab7-CA in either ISCs/EBs or ECs induced late endosome maturation (labeled by RAB7-GFP; forms large multivesicular bodies, see in the images below)[Figures for referees not shown.], which suggested the upregulation of endocytic activity.

Referee #3:

Du et al. present interesting findings related to lipoic acid, intestinal homeostasis and aging in *Drosophila*. There are many strengths to this manuscript, including the large amount of genetic experiments that provide mechanistic insight. The most exciting aspect of the study is that feeding ALA to aged flies can rejuvenate aspects of ISC aging. This is very interesting. On the whole, the experiments are carefully carried out, however, I do have some technical concerns and suggestions to improve the manuscript.

Response:

We appreciate that this reviewer finds our study original, important, and high quality. We believe the revisions we made as detailed below have addressed the concerns raised.

1) It is clear that ISC homeostasis can be improved by ALA feeding in 26day old flies. It would increase the impact of the paper if it were shown that day 26 feeding could prolong lifespan. In addition, multiple 'lab strains' should be tested. w1118 is a notoriously unhealthy line.

Response:

We appreciate these suggestions. We have performed the lifespan analyses by ALA feeding in 26-day old flies. It indeed showed a minor increase of the lifespan of flies (See Fig 3J and EV2J). We have fed ALA to Canton-S (CS, BDSC 64349; an often used wild-type line) and found that ALA feeding also extend the lifespan of CS flies (See Fig 3I and EV2I). We have included these new data in the revised manuscript.

2) It is critical to assay feeding behavior in control flies fed ALA. This is missing. If ALA impaired feeding in control 26d old flies that would be a confounding factor

Response:

We appreciated this suggestion. We have checked the feeding behavior (food intake) of flies at the 26th day. We found that the flies with or without ALA administration did not show a significant difference. So ALA itself does not impair or promote the feeding behavior of flies. We have included this new data in Fig EV2G in the revised manuscript.

3) It is great that the authors sought to examine autophagic activity in aged ISCs. However, my understanding is that the reagent that was used ATg8 tandem repeat is under UAS-control. Hence, in using this reagent aren't the authors overexpressing Atg8? If so, this is a confounding factor, no?

Response:

*Thanks for this question. In Figs 6F-H and 6O-P, we observed the autophagic activity that induced by starvation in ISCs/EBs using *esg-GAL4>UAS-mCherry-GFP-Atg8a* reporter. The *UAS-mCherry-GFP-Atg8a* line (BDSC: 37749) was widely used for observing the autophagic flux (autophagosomes and autolysosomes)[17-19]. overexpression of this *UAS-mCherry-GFP-Atg8a* allele has not yet been reported to stimulate autophagy (Maybe because this reporter allele is functional impaired by fusing with GFP and mCherry proteins at the amino terminus of ATG8a) [20]. In aged flies, *Atg8a* reduced in both mRNA and protein level[21], the *UAS-Atg8a^{EP362}* line (BDSC: 10107) was used to induce the overexpression of *Atg8a* for promoting the base levels of autophagy in ISCs/EBs. Overexpression of this *UAS-Atg8* line (BDSC: 10107) and another *UAS-Atg8* line (BDSC: 51656) in the nervous system has been reported to effectively promote basal levels of autophagy and enhance longevity in adult *Drosophila* [21].*

1. Wang C, Guo X, Dou K, Chen H, Xi R (2015) Ttk69 acts as a master repressor of enteroendocrine cell specification in *Drosophila* intestinal stem cell lineages. *Development* **142**:

3321-31

2. Park JS, Na HJ, Pyo JH, Jeon HJ, Kim YS, Yoo MA (2015) Requirement of ATR for maintenance of intestinal stem cells in aging *Drosophila*. *Aging (Albany NY)* **7**: 307-18
3. Koehler CL, Perkins GA, Ellisman MH, Jones DL (2017) Pink1 and Parkin regulate *Drosophila* intestinal stem cell proliferation during stress and aging. *J Cell Biol* **216**: 2315-2327
4. Loza-Coll MA, Southall TD, Sandall SL, Brand AH, Jones DL (2014) Regulation of *Drosophila* intestinal stem cell maintenance and differentiation by the transcription factor Escargot. *EMBO J* **33**: 2983-96
5. Lourenco FC, Munro J, Brown J, Cordero J, Stefanatos R, Strathdee K, Orange C, Feller SM, Sansom OJ, Vidal M, *et al.* (2014) Reduced LIMK2 expression in colorectal cancer reflects its role in limiting stem cell proliferation. *Gut* **63**: 480-93
6. Amcheslavsky A, Ito N, Jiang J, Ip YT (2011) Tuberous sclerosis complex and Myc coordinate the growth and division of *Drosophila* intestinal stem cells. *J Cell Biol* **193**: 695-710
7. Shaw RL, Kohlmaier A, Polesello C, Veelken C, Edgar BA, Tapon N (2010) The Hippo pathway regulates intestinal stem cell proliferation during *Drosophila* adult midgut regeneration. *Development* **137**: 4147-58
8. Jin Y, Patel PH, Kohlmaier A, Pavlovic B, Zhang C, Edgar BA (2017) Intestinal Stem Cell Pool Regulation in *Drosophila*. *Stem Cell Reports* **8**: 1479-1487
9. Singh SR, Zeng X, Zhao J, Liu Y, Hou G, Liu H, Hou SX (2016) The lipolysis pathway sustains normal and transformed stem cells in adult *Drosophila*. *Nature* **538**: 109-113
10. Stenmark H, Parton RG, Steele-Mortimer O, Lutcke A, Gruenberg J, Zerial M (1994) Inhibition of rab5 GTPase activity stimulates membrane fusion in endocytosis. *EMBO J* **13**:

1287-96

11. Li G, Stahl PD (1993) Structure-function relationship of the small GTPase rab5. *J Biol Chem* **268**: 24475-80
12. Xu C, Tang HW, Hung RJ, Hu Y, Ni X, Housden BE, Perrimon N (2019) The Septate Junction Protein Tsp2A Restricts Intestinal Stem Cell Activity via Endocytic Regulation of aPKC and Hippo Signaling. *Cell Rep* **26**: 670-688 e6
13. Wang S, Zhao Z, Rodal AA (2019) Higher-order assembly of Sorting Nexin 16 controls tubulation and distribution of neuronal endosomes. *J Cell Biol* **218**: 2600-2618
14. Lone M, Kungl T, Koper A, Bottenberg W, Kammerer R, Klein M, Sweeney ST, Auburn RP, O'Kane CJ, Prokop A (2010) The nuclear protein Waharan is required for endosomal-lysosomal trafficking in Drosophila. *J Cell Sci* **123**: 2369-74
15. Li B, Wong C, Gao SM, Zhang R, Sun R, Li Y, Song Y (2018) The retromer complex safeguards against neural progenitor-derived tumorigenesis by regulating Notch receptor trafficking. *Elife* **7**
16. Lorincz P, Toth S, Benko P, Lakatos Z, Boda A, Glatz G, Zobel M, Bisi S, Hegedus K, Takats S, *et al.* (2017) Rab2 promotes autophagic and endocytic lysosomal degradation. *J Cell Biol* **216**: 1937-1947
17. Jean S, Cox S, Nassari S, Kiger AA (2015) Starvation-induced MTMR13 and RAB21 activity regulates VAMP8 to promote autophagosome-lysosome fusion. *EMBO Rep* **16**: 297-311
18. Nakamura S, Oba M, Suzuki M, Takahashi A, Yamamuro T, Fujiwara M, Ikenaka K, Minami S, Tabata N, Yamamoto K, *et al.* (2019) Suppression of autophagic activity by Rubicon

is a signature of aging. *Nat Commun* **10**: 847

19. Nagy P, Varga A, Piracs K, Hegedus K, Juhasz G (2013) Myc-driven overgrowth requires unfolded protein response-mediated induction of autophagy and antioxidant responses in *Drosophila melanogaster*. *PLoS Genet* **9**: e1003664

20. Mauvezin C, Ayala C, Braden CR, Kim J, Neufeld TP (2014) Assays to monitor autophagy in *Drosophila*. *Methods* **68**: 134-9

21. Simonsen A, Cumming RC, Brech A, Isakson P, Schubert DR, Finley KD (2008) Promoting basal levels of autophagy in the nervous system enhances longevity and oxidant resistance in adult *Drosophila*. *Autophagy* **4**: 176-84

Dear Prof. Chen,

Thank you for submitting the revised version of your manuscript. It has now been seen by both of the original referees.

As you can see, the referees find that the study is significantly improved during revision and recommend publication. (Please note that referee #3 did not send a report, just notified us that he/she has no remaining concerns and recommends publication.) Before I can accept the manuscript, I need you to address some minor points below:

- We noted that Jiaqi Zhou and Xiaorong Li are missing from the Author Contributions section.
- We found that some of the figure callouts are potentially confusing - e.g. (comparing Fig 1F to 1E; see Figs 1G-H and EV1C). Please consider simplifying them.
- We noted that Fig 7C is currently not called out in the text.
- Please separate the EV source data to another folder.
- Figure 8 has many panels, please consider splitting it into two figures.
- Please convert the reagent table (Drosophila lines and genotypes, antibody list) as an EV Table. Please remember to call it out in the text accordingly.
- Papers published in EMBO Reports include a 'Synopsis' to further enhance discoverability. Synopses are displayed on the html version of the paper and are freely accessible to all readers. The synopsis includes a short standfirst summarizing the study in 1 or 2 sentences that summarize the key findings of the paper and are provided by the authors and streamlined by the handling editor. I would therefore ask you to include your synopsis blurb.
- In addition, please provide an image for the synopsis. This image should provide a rapid overview of the question addressed in the study but still needs to be kept fairly modest since the image size cannot exceed 550x400 pixels.
- Our production/data editors have asked you to clarify several points in the figure legends (see attached document). Please incorporate these changes in the attached word document and return it with track changes activated.

Thank you again for giving us to consider your manuscript for EMBO Reports, I look forward to your minor revision.

Kind regards,

Deniz Senyilmaz Tiebe

--

Deniz Senyilmaz Tiebe, PhD
Editor
EMBO Reports

Referee #1:

This is an excellent revision. The authors have thoroughly responded to the reviewers' comments, and added some new experiments that are quite impressive. These significantly strengthen the

model they present. For example, they performed LC-ESI-MS/MS analyses and measured the ALA levels in the midguts of flies during aging (Figs 1L-O, and Figs EV2A-E), and in the midguts of flies with Las depleted (Figs 2A-C). These data support that ALA is indeed downregulated in the midguts of flies upon aging, and in flies with LAS knockdown. Further, they performed Las-RNAi MARCM clones. These showed that LAS does not regulate ISC differentiation, but does regulate the rate of ISC mitotic proliferation. These experiments have improved this paper, which was already novel and interesting. In addition, the authors provide more quantitative data to support the representative pictures. This is another point for a good revision. Regarding the "too good to be true" data we were concerned about in the 1st round review, the authors now provide their raw data as supplemental tables, and show several different statistical analyses, which in general make sense. Although they still don't provide a clear mechanism for how ALA affects endocytosis and autophagy, in our opinion this is not an issue that should prevent publication in EMBO Reports. Overall, this paper is very coherent, having a complete genetic analysis and interesting, novel results that should be of interest to the field.

Below, we have provided our response (*italic maroon color*).

- We noted that Jiaqi Zhou and Xiaorong Li are missing from the Author Contributions section.

Response:

We have added Jiaqi Zhou and Xiaorong Li to the Author Contributions section in the revised manuscript.

- We found that some of the figure callouts are potentially confusing - e.g. (comparing Fig 1F to 1E; see Figs 1G-H and EV1C). Please consider simplifying them.

Response:

We have simplified these figure callouts in the revised manuscript.

- We noted that Fig 7C is currently not called out in the text.

Response:

We have called out Fig 7C in the revised manuscript.

- Please separate the EV source data to another folder.

Response:

We have separated the original source data zip file into two zip files. One named Source datasets (which contains the Excel files of statistic data for Figures), and the other one named EV source datasets (which contains the Excel files of statistic data for Figure EVs).

- Figure 8 has many panels, please consider splitting it into two figures.

Response:

We have split the original Figure 8 into two figures (now called Figure 8 and Figure 9 in the revised manuscript). The original Figure 9 now becomes Figure 10. Correspondingly, we re-labeled and claimed these figure callouts in the revised text, Figures and Figure legends.

- Please convert the reagent table (Drosophila lines and genotypes, antibody

list) as an EV Table. Please remember to call it out in the text accordingly.

Response:

We have converted the original reagent table (Drosophila lines and genotypes, antibody list) into three EV Tables (named Table EV3, Table EV4, and Table EV5) and uploaded them as reagent tables in your online submission system. We also have called them out in the revised Materials and Methods section.

- Papers published in EMBO Reports include a 'Synopsis' to further enhance discoverability. Synopses are displayed on the html version of the paper and are freely accessible to all readers. The synopsis includes a short standfirst summarizing the study in 1 or 2 sentences that summarize the key findings of the paper and are provided by the authors and streamlined by the handling editor. I would therefore ask you to include your synopsis blurb.

Response:

We have provided a word file of our Synopsis blurb in your online submission system.

- In addition, please provide an image for the synopsis. This image should provide a rapid overview of the question addressed in the study but still needs to be kept fairly modest since the image size cannot exceed 550x400 pixels.

Response:

We have provided an image of our Synopsis with tiff format in your online submission system.

- Our production/data editors have asked you to clarify several points in the figure legends (see attached document). Please incorporate these changes in the attached word document and return it with track changes activated.

Response:

We have noticed these comments added by the production/data editors. We have incorporated modifications and changes in our revised manuscript text and uploaded a version of our revised manuscript text (without track changes) through your online submission system.

I also returned you a version of our manuscript text with track changes activated and the comments added by the production/data editor through my e-mail (in the attachment).

Dear Prof. Chen,

Thank you for submitting your revised manuscript to EMBO Reports. I have now looked at everything and all looks fine. Therefore I am very pleased to accept your manuscript for publication in EMBO Reports.

Congratulations on a nice work!

Kind regards,

Deniz

--

Deniz Senyilmaz Tiebe, PhD

Editor

EMBO Reports

**

At the end of this email I include important information about how to proceed. Please ensure that you take the time to read the information and complete and return the necessary forms to allow us to publish your manuscript as quickly as possible.

As part of the EMBO publication's Transparent Editorial Process, EMBO reports publishes online a Review Process File to accompany accepted manuscripts. As you are aware, this File will be published in conjunction with your paper and will include the referee reports, your point-by-point response and all pertinent correspondence relating to the manuscript.

If you do NOT want this File to be published, please inform the editorial office within 2 days, if you have not done so already, otherwise the File will be published by default [contact: emboreports@embo.org]. If you do opt out, the Review Process File link will point to the following statement: "No Review Process File is available with this article, as the authors have chosen not to make the review process public in this case."

Should you be planning a Press Release on your article, please get in contact with emboreports@wiley.com as early as possible, in order to coordinate publication and release dates.

Thank you again for your contribution to EMBO reports and congratulations on a successful publication. Please consider us again in the future for your most exciting work.

THINGS TO DO NOW:

You will receive proofs by e-mail approximately 2-3 weeks after all relevant files have been sent to our Production Office; you should return your corrections within 2 days of receiving the proofs.

Please inform us if there is likely to be any difficulty in reaching you at the above address at that time. Failure to meet our deadlines may result in a delay of publication, or publication without your

corrections.

All further communications concerning your paper should quote reference number EMBOR-2019-49583V3 and be addressed to emboreports@wiley.com.

Should you be planning a Press Release on your article, please get in contact with emboreports@wiley.com as early as possible, in order to coordinate publication and release dates.

Corresponding Author Name: Haiyang Chen

Manuscript Number: EMBOR-2019-49583V2